# Dithiothreitol causes toxicity in *C. elegans* by modulating the methionine–homocysteine cycle

**Gokul G[1], Jogender Singh[2]***

[1]Department of Biological Sciences, Indian Institute of Science Education and Research, Bhopal, Bhopal, India; [2]Department of Biological Sciences, Indian Institute of Science Education and Research, Mohali, Mohali, India

**Abstract** The redox reagent dithiothreitol (DTT) causes stress in the endoplasmic reticulum (ER) by disrupting its oxidative protein folding environment, which results in the accumulation and misfolding of the newly synthesized proteins. DTT may potentially impact cellular physiology by ER-independent mechanisms; however, such mechanisms remain poorly characterized. Using the nematode model *Caenorhabditis elegans*, here we show that DTT toxicity is modulated by the bacterial diet. Specifically, the dietary component vitamin B12 alleviates DTT toxicity in a methionine synthase-dependent manner. Using a forward genetic screen, we discover that loss-of-function of R08E5.3, an *S*-adenosylmethionine (SAM)-dependent methyltransferase, confers DTT resistance. DTT upregulates R08E5.3 expression and modulates the activity of the methionine–homocysteine cycle. Employing genetic and biochemical studies, we establish that DTT toxicity is a result of the depletion of SAM. Finally, we show that a functional IRE-1/XBP-1 unfolded protein response pathway is required to counteract toxicity at high, but not low, DTT concentrations.

## Editor's evaluation

DTT is thought to lead to cellular stress through its effect on the ER milieu and on disulfide bond formation in the ER. The authors show that DTT also affects the methionine cycle and that SAM depletion in fact leads to significant DTT toxicity in *C. elegans*, challenging the current model.

*For correspondence:
jogender@iisermohali.ac.in

**Competing interest:** The authors declare that no competing interests exist.

## Introduction

Diverse biological processes occur via oxidation–reduction (redox) reactions, and therefore, a redox balance is essential for cellular and organismal homeostasis (*Trachootham et al., 2008*). A consequence of biological redox reactions is the generation of reactive oxygen species (ROS). The physiological flux of ROS regulates cellular processes essential for cell survival, maintenance, and aging (*Sies and Jones, 2020*; *Trachootham et al., 2008*). In contrast, excessive ROS leads to oxidative stress, a hallmark of many pathological states (*Sies and Jones, 2020*). To combat excessive ROS, the cellular systems have developed antioxidant networks, including thioredoxin, glutathione, glutathione peroxidases, catalase, and superoxide dismutases (*Kurutas, 2016*). In addition to endogenous antioxidants, dietary antioxidant supplements are taken under healthy as well as pathological conditions to counteract ROS (*Kurutas, 2016*). Animal and cell model studies suggest that thiol-based antioxidants, including glutathione, *N*-acetylcysteine (NAC), and dithiothreitol (DTT), may attenuate ROS-induced pathological conditions such as hepatic, hematopoietic, intestinal, and renal injuries (*Heidari et al., 2018*; *Heidari et al., 2016b*; *Li et al., 2019*), Alzheimer's disease (*Pocernich and Butterfield, 2012*), and dopamine-induced cell death (*Offen et al., 1996*). While the thiol-based antioxidants appear

**eLife digest** Animal and plant cells synthesize a significant fraction of their proteins on a structure known as the endoplasmic reticulum. Researchers often use the molecule dithiothreitol to specifically target this compartment and learn more about its role. The toxin works by disturbing the complex chemical environment present in the reticulum, which is required for the proteins to assemble properly. However, it is important to clarify whether dithiothreitol could also affect other parts of the cell, as this could give rise to misleading results.

To explore this possibility, Gokul G and Jogender Singh studied the effects of dithiothreitol on the millimetre-long roundworm *Caenorhabditis elegans*. Their experiments revealed that vitamin B12 could protect against dithiothreitol toxicity via a complex cascade of molecular events which reduced the levels of an important regulatory molecule known as S-adenosylmethionine. Crucially, the chemical reactions that dithiothreitol targeted took place outside the reticulum, suggesting that the toxin impairs processes in the wider cell.

These results suggest that dithiothreitol should be reconsidered for use in endoplasmic reticulum studies. However, they also imply that this toxin could be beneficial in small doses, as a reduced concentration of S-adenosylmethionine increases lifespan and health in a variety of organisms.

to improve various pathological conditions, their broad effects on cellular physiology remain poorly characterized. Recently, it was shown that glutathione and NAC could have detrimental effects on the health and lifespan of *Caenorhabditis elegans* (*Gusarov et al., 2021*). Therefore, a better characterization of the physiological effects of various thiol antioxidants is necessary to harness their therapeutic value.

DTT is a commonly used redox reagent that contains two thiol groups (*Cleland, 1964*) and is known to reduce protein disulfide bonds (*Konigsberg, 1972*). Because the endoplasmic reticulum (ER) has an oxidative environment conducive to disulfide bond formation, DTT reduces disulfide bonds in the ER (*Braakman et al., 1992*). Therefore, DTT is widely used as an ER-specific stressor (*Braakman et al., 1992*; *Oslowski and Urano, 2011*). While DTT has been shown to improve several ROS-based pathological conditions (*Heidari et al., 2018*; *Heidari et al., 2016b*; *Li et al., 2019*; *Offen et al., 1996*), high amounts of DTT are toxic and lead to cell death which is thought to be mediated by increased ER stress (*Li et al., 2011*; *Qin et al., 2010*; *Xiang et al., 2016*). However, some studies suggest that DTT may induce apoptosis by generating hydrogen peroxide and oxidative stress (*Held et al., 1996*; *Held and Melder, 1987*). Similarly, several other DTT-mediated phenotypes are shown to be independent of ER stress (*Guillemette et al., 2007*; *MacKenzie et al., 2005*; *Messias Sandes et al., 2019*; *Tartier et al., 2000*). Therefore, how DTT impacts cellular physiology remains to be fully understood.

Here, using the *C. elegans* model, we characterized the physiological effects of DTT. We showed that DTT causes developmental defects in *C. elegans* in a bacterial diet-dependent manner. The dietary component vitamin B12 alleviated development defects caused by DTT by acting as a cofactor for methionine synthase. To understand the interplay between DTT and vitamin B12, we isolated *C. elegans* mutants in a forward genetic screen that were resistant to DTT toxicity. Loss-of-function of R08E5.3, an *S*-adenosylmethionine (SAM)-dependent methyltransferase, imparted DTT resistance. DTT resulted in the upregulation of R08E5.3 expression and, therefore, modulated the methionine–homocysteine cycle. DTT resulted in the depletion of SAM, and supplementation of methionine and choline could rescue DTT toxicity. Modulation of the methionine–homocysteine cycle by DTT also resulted in the upregulation of the ER and mitochondrial unfolded protein response (UPR) pathways. Finally, we showed that while DTT toxicity primarily occurred via the modulation of the methionine–homocysteine cycle, a functional IRE-1/XBP-1 UPR pathway was required to counteract DTT toxicity.

## Results

### DTT affects *C. elegans* development in a bacterial diet-dependent manner

DTT is known to be toxic to *C. elegans* and affects its development (*Kozlowski et al., 2014*). We first studied *C. elegans* development retardation at different concentrations of DTT. In the presence of

*Escherichia coli* OP50 diet, *C. elegans* was at the L1 or L2 stage at 5 mM and higher DTT concentrations at 72 hr of hatching, while the control animals without DTT had developed to adulthood at the same time (*Figure 1A, B*). We also studied development retardation in the presence of DTT on *E. coli* HT115 bacterium, a strain commonly used for feeding-based RNA interference (RNAi). Surprisingly, we observed that *C. elegans* developed much better on DTT in the presence of *E. coli* HT115 diet than *E. coli* OP50 diet (*Figure 1A–C*). These results suggested that some bacterial components might be able to modulate DTT toxicity.

To gain an understanding of the potential bacterial components that might modulate DTT toxicity, we studied the development of *C. elegans* in the presence of DTT on different bacterial diets, including *Comamonas aquatica* DA1877, *Pseudomonas aeruginosa* PA14 *gacA* mutant, *Salmonella enterica*, and *Serratia marcescens*. Because *P. aeruginosa* virulence might affect the development of *C. elegans*, we used the *gacA* mutant, which has drastically reduced virulence (*Tan et al., 1999*). Interestingly, *C. elegans* showed no developmental defects on *C. aquatica* DA1877 and *P. aeruginosa* PA14 diets in the presence of 10 mM DTT (*Figure 1D, E*). Similarly, *C. elegans* developed much better on *S. enterica* and *S. marcescens* diets compared to the *E. coli* OP50 diet at lower DTT concentrations (*Figure 1—figure supplement 1A–D*). These results showed that bacterial diet has a major effect on DTT toxicity, and likely some bacterial diet component(s) modulated DTT toxicity.

Previous studies have shown that relative to the *E. coli* OP50 diet, a *C. aquatica* DA1877 diet accelerates *C. elegans* development primarily because it contains higher levels of vitamin B12 (*Watson et al., 2014*). Similarly, *E. coli* HT115 and *P. aeruginosa* PA14 diets are known to have higher amounts of vitamin B12 than the *E. coli* OP50 diet (*Revtovich et al., 2019*; *Watson et al., 2014*). The amount of dietary vitamin B12 available to *C. elegans* can be reliably determined by a dietary sensor strain expressing green fluorescent protein (GFP) under the promoter of acyl-coenzyme A dehydrogenase (*acdh-1*) gene (*Watson et al., 2014*). Low levels of vitamin B12 result in the accumulation of the short-chain fatty acid propionate, which leads to transcriptional activation of *acdh-1* (*Watson et al., 2014*). Thus, the GFP levels in the *acdh-1p::GFP* animals indirectly report the levels of dietary vitamin B12 available to *C. elegans*. When the animals experience low levels of vitamin B12, *acdh-1p::GFP* is strongly induced, and the level of GFP expression reduces as the animals obtain increasing levels of vitamin B12. Using *acdh-1p::GFP* animals, we first tested the levels of vitamin B12 in the animals on different bacterial diets compared to the *E. coli* OP50 diet. As shown in *Figure 1F* and reported previously (*Watson et al., 2014*), the animals show very bright GFP expression on the *E. coli* OP50 diet, while the GFP levels were diminished on *E. coli* HT115, *S. enterica*, and *S. marcescens* diets. On the other hand, the animals showed no GFP expression on *C. aquatica* DA1877 and *P. aeruginosa* PA14 diets. The levels of GFP expression in *acdh-1p::GFP* animals on different bacterial diets correlated perfectly with the developmental defects on these bacterial diets in the presence of DTT (*Figure 1G*).

## Vitamin B12 alleviates DTT toxicity via methionine synthase

The correlation between the GFP levels in *acdh-1p::GFP* animals and the developmental defects of animals in the presence of DTT on different bacterial diets suggested that vitamin B12 could be the microbial component that alleviates DTT toxicity in *C. elegans*. To test this, we studied the development of N2 animals on *E. coli* OP50 diet containing 10 mM DTT and supplemented with 50 nM vitamin B12. In contrast to the development retardation on *E. coli* OP50 diet containing 10 mM DTT, the animals developed to adulthood on *E. coli* OP50 diet containing 10 mM DTT supplemented with 50 nM vitamin B12 (*Figure 2A, B*). We further asked whether the DTT-mediated development retardation could be reversed by supplementation of vitamin B12. To this end, vitamin B12 was added to the L1-arrested animals after 72 hr of hatching on *E. coli* OP50 plates containing 10 mM DTT. After 48 hr of incubation post vitamin B12 supplementation, more than 95% of the animals developed to adulthood (*Figure 2—figure supplement 1A, B*). On the other hand, the control animals on *E. coli* OP50 plates containing 10 mM DTT, but without vitamin B12 supplementation, were still at the L1–L2 stages after 120 hr of hatching. Taken together, these studies showed that vitamin B12 is sufficient to alleviate and revert the toxic effects of DTT.

Vitamin B12 is an essential cofactor for two metabolic enzymes: methylmalonyl-CoA mutase (MMCM-1 in *C. elegans*), which converts methylmalonyl-CoA to succinyl-CoA in the propionyl-CoA breakdown pathway, and methionine synthase (METR-1 in *C. elegans*), which converts homocysteine to methionine in the methionine–homocysteine cycle. We asked whether one or both of these enzymes

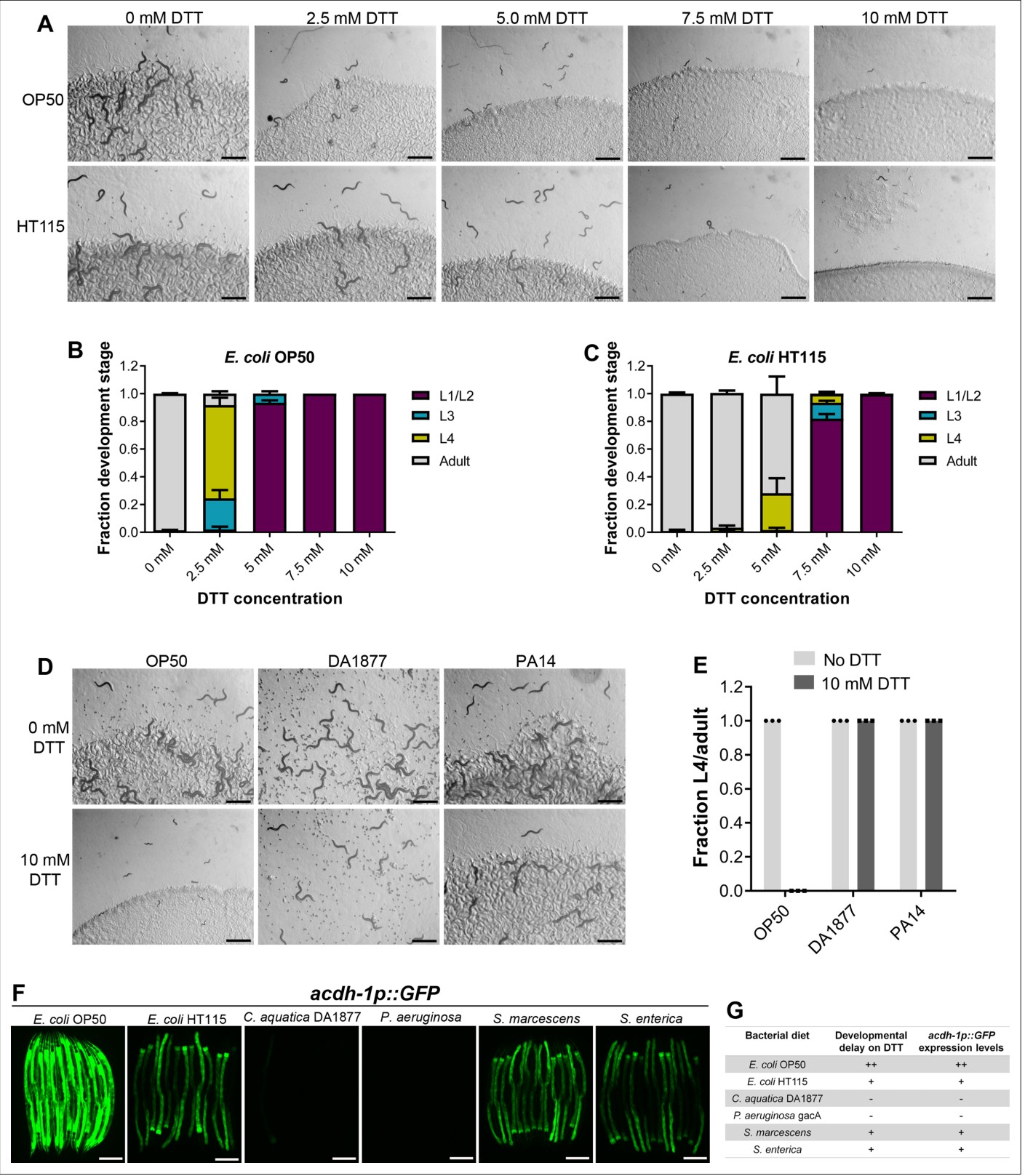

**Figure 1.** Dithiothreitol (DTT) affects *C. elegans* development in a diet-dependent manner. (**A**) Representative images of wild-type N2 *C. elegans* on various concentrations of DTT on *E. coli* OP50 and *E. coli* HT115 diets after 72 hr of hatching at 20°C. Scale bar = 1 mm. Quantification of different developmental stages of wild-type N2 *C. elegans* on various concentrations of DTT on *E. coli* OP50 (**B**) and *E. coli* HT115 (**C**) diets after 72 hr of hatching at 20°C (*n* = 3 biological replicates; animals per condition per replicate >80). (**D**) Representative images of wild-type N2 *C. elegans* after 72 hr of

*Figure 1 continued on next page*

*Figure 1 continued*

hatching at 20°C on *E. coli* OP50, *C. aquatica* DA1877, and *P. aeruginosa* PA14 *gacA* mutant diets containing either 0 or 10 mM DTT. Scale bar = 1 mm. (**E**) Fraction L4 or adult wild-type N2 *C. elegans* after 72 hr of hatching at 20°C on *E. coli* OP50, *C. aquatica* DA1877, and *P. aeruginosa* PA14 *gacA* mutant diets containing either 0 or 10 mM DTT (*n* = 3 biological replicates; animals per condition per replicate >80). (**F**) Representative fluorescence images of *acdh-1p::GFP* animals (**Watson et al., 2014**) grown on various bacterial diets. Scale bar = 200 μm. (**G**) Table summarizing the effects of bacterial diet on DTT-induced developmental delay and on GFP levels in *acdh-1p::GFP* animals.

The online version of this article includes the following source data and figure supplement(s) for figure 1:

**Source data 1.** DTT affects *C. elegans* development in a diet-dependent manner.

**Figure supplement 1.** Dithiothreitol (DTT) affects *C. elegans* development in a diet-dependent manner.

**Figure supplement 1—source data 1.** Dithiothreitol (DTT) affects *C. elegans* development in a diet-dependent manner.

are required for alleviating DTT toxicity. While vitamin B12 supplementation rescued the toxic effects of DTT in *mmcm-1(ok1637)* animals, it failed to do so in *metr-1(ok521)* animals (**Figure 2C, D**). The *metr-1(ok521)* animals were at the L1 stage on *E. coli* OP50 diet containing 10 mM DTT regardless of vitamin B12 supplementation (**Figure 2C**). Further, *mmcm-1(ok1637)* animals did not show any significant development retardation effects when grown on *C. aquatica* DA1877 and *P. aeruginosa* PA14 diets containing 10 mM DTT (**Figure 2E, F**). On the other hand, *metr-1(ok521)* animals failed to develop when grown on *C. aquatica* DA1877 and *P. aeruginosa* PA14 diets containing 10 mM DTT (**Figure 2E, F**). Taken together, these studies showed that vitamin B12 could alleviate DTT toxicity via the methionine synthase enzyme.

## DTT toxicity is mediated via a SAM-dependent methyltransferase

To understand why DTT toxicity depended on vitamin B12 and the methionine–homocysteine cycle, we carried out a forward genetic screen for mutants that would develop to L4/adult stage at 72 hr posthatching on *E. coli* OP50 diet containing 10 mM DTT. From a screen of approximately 50,000 ethyl methanesulfonate (EMS)-mutagenized haploid genomes, 12 mutants were isolated that exhibited DTT resistance and developed on *E. coli* OP50 diet containing 10 mM DTT (**Figure 3—figure supplement 1A, B**). To identify the causative mutations, we conducted the whole-genome sequencing of all the 12 mutants after backcrossing them six times with the wild-type N2 strain. The sequenced genomes were aligned with the reference genome of *C. elegans*. After subtraction of the common variants, linkage maps of single-nucleotide polymorphisms (SNPs) were obtained (**Figure 3—figure supplement 2**). All of the mutants had linkage to chromosome V. Analysis of the protein-coding genes carrying mutations in the linked regions of each mutant revealed that all of the mutants had a mutation in the gene R08E5.3 (**Figure 3A, B**), which was recently named as *rips-1* (*R*hy-1-*I*nteracting *P*rotein in *S*ulfide response) (personal communication with Wormbase). *rips-1* is predicted to encode a protein with a SAM-dependent methyltransferase domain. SAM-dependent methyltransferases catalyze diverse methylation reactions where a methyl group is transferred to the substrate from SAM, resulting in the conversion of SAM to *S*-adenosylhomocysteine (SAH) in the methionine–homocysteine cycle. Fortuitously, the allele *gk231506* available in the *Caenorhabditis* Genetics Center (CGC) has the same molecular change in the protein RIPS-1 as the allele *jsn6* obtained in our screen has. *rips-1(gk231506)* animals showed DTT resistance and developed on *E. coli* OP50 diet containing 10 mM DTT (**Figure 3C, D**).

Three of the isolated alleles of *rips-1*, *jsn4*, *jsn11*, and *jsn12* had premature stop codon mutations and one allele (*jsn8*) had a splice acceptor mutation (**Figure 3A**). All of these mutations are expected to result in a loss-of-function of the gene. To confirm that resistance to DTT toxicity is imparted by loss-of-function mutations in *rips-1*, we created *rips-1(jsn11)* animals expressing *rips-1* under its own promoter. As shown in **Figure 3E**, the DTT-resistant phenotype of *rips-1(jsn11)* animals was completely rescued in these animals. We further determined the expression pattern of *rips-1* by expressing GFP driven by its own 918 bp promoter in wild-type N2 animals. As shown in **Figure 3F**, *rips-1* is expressed in the intestine.

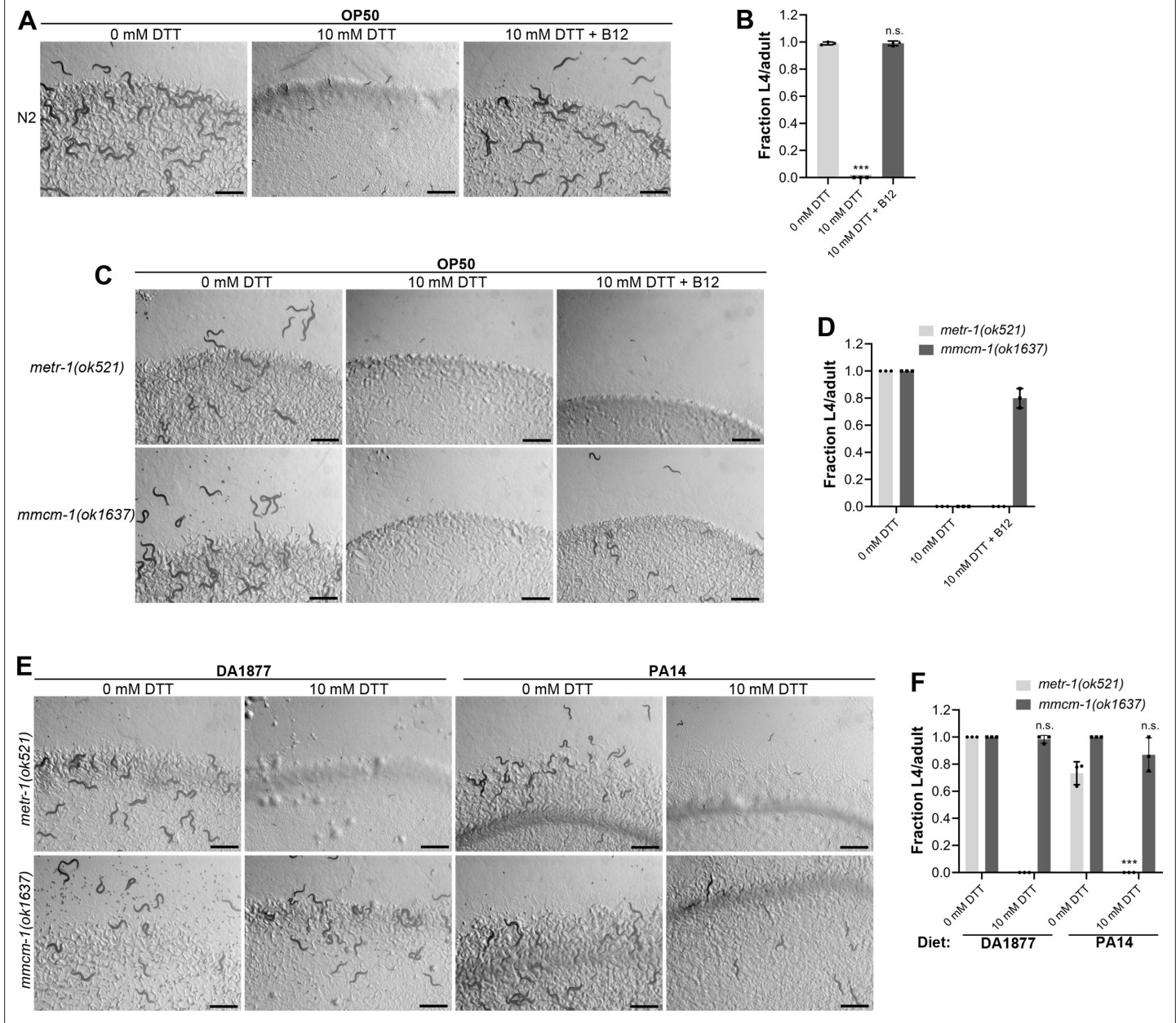

**Figure 2.** Vitamin B12 alleviates dithiothreitol (DTT) toxicity via methionine synthase. (**A**) Representative images of wild-type N2 *C. elegans* after 72 hr of hatching at 20°C on *E. coli* OP50 diet containing 0, 10, and 10 mM DTT supplemented with 50 nM vitamin B12. Scale bar = 1 mm. (**B**) Fraction L4 or adult wild-type N2 *C. elegans* after 72 hr of hatching at 20°C on *E. coli* OP50 diet containing 0, 10, and 10 mM DTT supplemented with 50 nM vitamin B12. p values are relative to 0 mM DTT. ***p < 0.001 via the *t*-test. n.s., nonsignificant (*n* = 3 biological replicates; animals per condition per replicate >80). (**C**) Representative images of *metr-1(ok521)* and *mmcm-1(ok1637)* animals after 72 hr of hatching at 20°C on *E. coli* OP50 diet containing 0 mM DTT, 10 mM DTT, and 10 mM DTT supplemented with 50 nM vitamin B12. Scale bar = 1 mm. (**D**) Fraction L4 or adult *metr-1(ok521)* and *mmcm-1(ok1637)* animals after 72 hr of hatching at 20°C on *E. coli* OP50 diet containing 0, 10, and 10 mM DTT supplemented with 50 nM vitamin B12 (*n* = 3 biological replicates; animals per condition per replicate >60). (**E**) Representative images of *metr-1(ok521)* and *mmcm-1(ok1637)* animals after 72 hr of hatching at 20°C on *C. aquatica* DA1877 and *P. aeruginosa* PA14 *gacA* mutant diets containing either 0 or 10 mM DTT. Scale bar = 1 mm. (**F**) Fraction L4 or adult *metr-1(ok521)* and *mmcm-1(ok1637)* animals after 72 hr of hatching at 20°C on *C. aquatica* DA1877 and *P. aeruginosa* PA14 *gacA* mutant diets containing either 0 or 10 mM DTT. p values are relative to the corresponding 0 mM DTT condition for each mutant. ***p < 0.001 via the *t*-test. n.s., nonsignificant (*n* = 3 biological replicates; animals per condition per replicate>60).

The online version of this article includes the following source data and figure supplement(s) for figure 2:

**Source data 1.** Vitamin B12 alleviates dithiothreitol (DTT) toxicity via methionine synthase.

**Figure supplement 1.** Vitamin B12 supplementation reverses dithiothreitol (DTT) toxicity.

**Figure supplement 1—source data 1.** Vitamin B12 supplementation reverses dithiothreitol (DTT) toxicity.

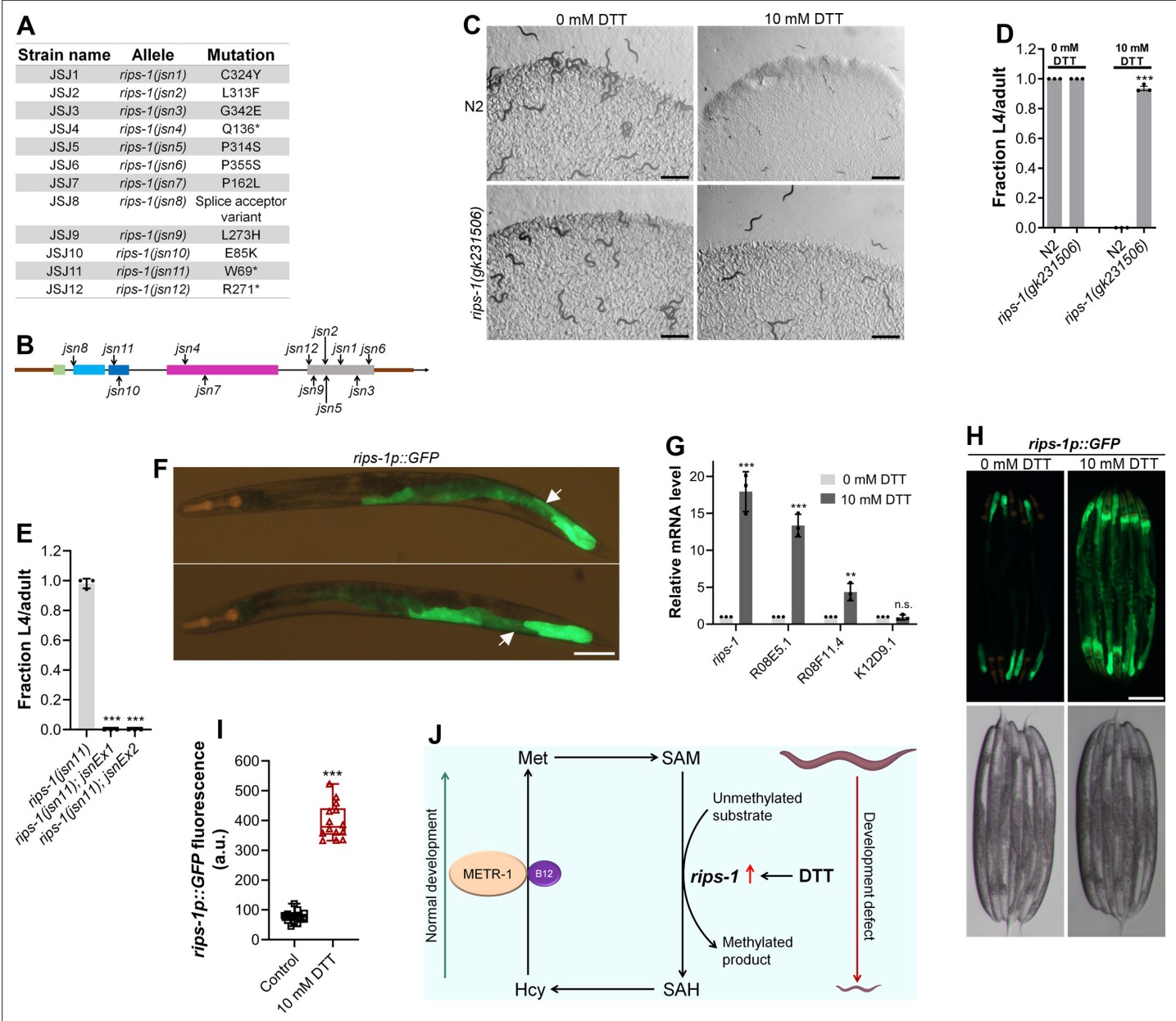

**Figure 3.** Dithiothreitol (DTT) causes developmental toxicity via *rips-1*. (**A**) Table summarizing the alleles of *rips-1* identified by whole-genome sequencing in different DTT-resistant strains. The corresponding amino acid changes in the RIPS-1 protein are also shown. (**B**) Mapping of the *rips-1* alleles identified in the forward genetic screen. (**C**) Representative images of N2 and *rips-1(gk231506)* animals after 84 hr of hatching at 20°C on *E. coli* OP50 diets containing either 0 or 10 mM DTT. Scale bar = 1 mm. (**D**) Fraction L4 or adult N2 and *rips-1(gk231506)* animals after 84 hr of hatching at 20°C on *E. coli* OP50 diets containing either 0 or 10 mM DTT. ***p < 0.001 via the *t*-test (n = 3 biological replicates; animals per condition per replicate >50). (**E**) Fraction L4 or adult *rips-1(jsn11)*, *rips-1(jsn11);jsnEx1*, and *rips-1(jsn11);jsnEx2* animals after 72 hr of hatching at 20°C on *E. coli* OP50 diet containing 10 mM DTT. *jsnEx1* and *jsnEx2* represent two independent extrachromosomal arrays containing *rips-1p::rips-1::SL2::GFP* and *myo-2p::mCherry*. ***p < 0.001 via the *t*-test (n = 3 biological replicates; animals per condition per replicate >50). (**F**) Representative fluorescence images of *jsnIs1[rips-1p::GFP +myo-2p::mCherry]* animals. The white arrows point at the intestinal regions showing GFP expression. Scale bar = 100 µm. (**G**) Gene expression analysis of N2 animals grown on *E. coli* OP50 diet containing 0 mM DTT until the young adult stage, followed by incubation on *E. coli* OP50 diet containing 0 or 10 mM DTT for 4 hr. ***p < 0.001 and **p < 0.01 via the *t*-test. n.s., nonsignificant (n = 3 biological replicates). (**H**) Representative fluorescence (top) and the corresponding brightfield (bottom) images of *rips-1p::GFP* animals grown on *E. coli* OP50 diet containing 0 mM DTT until the young adult stage, followed by incubation on *E. coli* OP50 diet containing 0 mM or 10 mM DTT for 8 hr. Scale bar = 200 µm. (**I**) Quantification of GFP levels of *rips-1p::GFP* animals grown on *E. coli* OP50 diet containing 0 mM DTT until the young adult stage, followed by incubation on *E. coli* OP50 diet containing 0 or 10 mM DTT for 8 hr. ***p < 0.001 via the *t*-test (n = 14 worms each). (**J**) Model depicting the effects of DTT and vitamin B12 on *C. elegans* development via the methionine–homocysteine cycle.

*Figure 3 continued on next page*

*Figure 3 continued*

The online version of this article includes the following source data and figure supplement(s) for figure 3:

**Source data 1.** Dithiothreitol (DTT) causes developmental toxicity via *rips-1*.

**Figure supplement 1.** Forward genetic screen resulted in the isolation of 12 dithiothreitol (DTT)-resistant mutants.

**Figure supplement 2.** Mapping of the mutations by whole-genome sequencing.

**Figure supplement 2—source data 1.** Mapping of the mutations by whole-genome sequencing.

**Figure supplement 3.** Dithiothreitol (DTT) causes developmental toxicity via *rips-1*.

**Figure supplement 3—source data 1.** Dithiothreitol (DTT) causes developmental toxicity via *rips-1*.

**Figure supplement 4.** Dithiothreitol (DTT) upregulates the mitochondrial unfolded protein response (UPR).

**Figure supplement 4—source data 1.** Dithiothreitol (DTT) upregulates the mitochondrial unfolded protein response (UPR).

## DTT modulates the methionine–homocysteine cycle by upregulating *rips-1* expression

We next asked how DTT exerted its toxic effects via *rips-1*. In the fungus *Aspergillus niger*, DTT is known to highly upregulate two SAM-dependent methyltransferases (*MacKenzie et al., 2005*). We asked whether DTT also upregulated the SAM-dependent methyltransferase *rips-1*, and any other closely related methyltransferases. *rips-1* shares high sequence similarity with three other SAM-dependent methyltransferases in *C. elegans*, R08E5.1 (BLAST *e* value: 1*e*−168), R08F11.4 (BLAST *e* value: 1*e*−166), and K12D9.1 (BLAST *e* value: 2*e*−152). Quantitative reverse transcription-PCR (qRT-PCR) analysis showed that DTT resulted in significant upregulation of the transcript levels of *rips-1*, R08E5.1, and R08F11.4 (*Figure 3G*). The enhanced expression of *rips-1* was confirmed in a reporter strain expressing GFP under the promoter of *rips-1* (*Figure 3H, I*).

Because DTT resulted in the upregulation of methyltransferases closely related to *rips-1*, we used RNAi knockdown to study the role of these methyltransferases in the toxic effects of DTT. As expected, knockdown of *rips-1* in the N2 animals resulted in a drastic improvement in their development in the presence of DTT (*Figure 3—figure supplement 3A, B*). Interestingly, knockdown of R08E5.1 resulted in a phenotype similar to *rips-1* knockdown. On the other hand, knockdown of R08F11.4 and K12D9.1 improved the development in the presence of DTT only marginally (*Figure 3—figure supplement 3A, B*). These results suggest that the *rips-1*-related methyltransferases might also be required for DTT toxicity. It is also possible that the knockdown of these methyltransferases results in the knockdown of *rips-1* by cross RNAi. The level of sequence similarity between *rips-1* and R08E5.1 (78 % identical) indicates that these genes would undergo complete cross RNAi (*Rual et al., 2007*). Because we did not recover mutants of any of the *rips-1*-related methyltransferases in our genetic screen in contrast to the 12 alleles of *rips-1*, the *rips-1*-related methyltransferases likely have only minor roles in DTT toxicity. However, our studies do not rule out the involvement of the *rips-1*-related methyltransferases in DTT toxicity.

To establish that the toxic effects of DTT are indeed due to the upregulation of *rips-1*, we tested the effects of overexpression of *rips-1* on DTT sensitivity. Animals overexpressing *rips-1* exhibited enhanced sensitivity to DTT toxicity (*Figure 3—figure supplement 3C, D*). Taken together, these results suggested that by increasing the expression of *rips-1*, DTT modulates the methionine–homocysteine cycle resulting in toxic effects that can be reversed by supplementation of vitamin B12 (*Figure 3J*).

Modulation of the methionine–homocysteine cycle or low dietary vitamin B12 are known to upregulate the mitochondrial UPR (*Amin et al., 2020*; *Wei and Ruvkun, 2020*). Since DTT modulates the methionine–homocysteine cycle and requires vitamin B12 to reverse those effects, we asked whether DTT can also lead to the upregulation of the mitochondrial UPR. To this end, we exposed the adult *hsp-6p::GFP* animals to DTT. Compared to the control animals, animals exposed to 10 mM DTT had upregulation of the mitochondrial UPR (*Figure 3—figure supplement 4A, B*). Supplementation of vitamin B12 rescued the upregulation of mitochondrial UPR by DTT, indicating that DTT-induced mitochondrial UPR via the methionine–homocysteine cycle.

## β-Mercaptoethanol, but not NAC, shares toxicity pathway with DTT

We tested whether the other thiol reagents, β-mercaptoethanol (β-ME) and NAC, caused toxicity via *rips-1*. Interestingly, *rips-1(jsn11)* animals exhibited resistance to β-ME toxicity, but not NAC toxicity (*Figure 4A, B*). We studied the effect of these thiol reagents on the expression levels of *rips-1p::GFP*. While NAC exposure did not affect the expression of *rips-1p::GFP*, β-ME resulted in its dramatic upregulation (*Figure 4C, D*). Finally, we studied the effects of vitamin B12 supplementation of the toxicities of β-ME and NAC. Vitamin B12 supplementation alleviated β-ME, but not NAC, toxicity (*Figure 4E, F*). These results suggested that β-ME shares the toxicity mechanism with DTT. On the other hand, NAC causes toxicity by a mechanism different from DTT.

## DTT toxicity is a result of SAM depletion

Next, we asked how perturbations of the methionine–homocysteine cycle activity led to DTT toxicity. Since DTT exerts its toxicity via the SAM-dependent methyltransferase RIPS-1, we postulated that either depletion of the metabolites upstream of RIPS-1 activity (methionine and SAM) or accumulation of metabolites downstream of RIPS-1 activity (SAH and homocysteine) or a combination of both would result in DTT toxicity. To discriminate between these possibilities, we carried out methionine supplementation assays in the presence of DTT. We reasoned that if DTT toxicity is because of methionine and/or SAM depletion, supplementation of methionine should attenuate DTT toxicity. On the other hand, if DTT toxicity is caused by the accumulation of SAH and/or homocysteine, supplementation of methionine should enhance DTT toxicity as methionine supplementation would likely enhance levels of SAH and/or homocysteine. The wild-type N2 animals had improved development when 10 mM DTT containing *E. coli* OP50 diet was supplemented with varying concentrations of methionine (*Figure 5A, B*). Methionine supplementation alleviated DTT toxicity in a dose-dependent manner (*Figure 5A, B*). We observed that *metr-1(ok521)* animals that failed to develop upon vitamin B12 supplementation in the presence of DTT (*Figure 2C, D*), developed upon supplementation of methionine (*Figure 5A, C*). These results suggested that depletion of methionine and/or SAM is the primary cause of DTT toxicity.

Next, we studied whether DTT toxicity was an outcome of depletion of methionine per se or its downstream metabolite SAM. The first step in the methionine–homocysteine cycle involves the conversion of methionine into SAM by methionine adenosyltransferase (*S*-adenosylmethionine synthetase or SAMS in *C. elegans*). Animals having knockdown of *sams-1* have defects in methionine to SAM conversion and exhibit significantly reduced levels of SAM (*Walker et al., 2011*). Because *sams-1* mutants have reduced conversion of methionine to SAM, supplementation of methionine would alleviate DTT toxicity in *sams-1* mutants only if DTT toxicity is caused by depletion of methionine and not SAM. Supplementation of methionine did not improve the development of *sams-1(ok2946)* animals on *E. coli* OP50 diet containing 10 mM DTT (*Figure 5D, E*). Further, *sams-1(ok2946)* animals also failed to develop on 10 mM DTT containing *E. coli* OP50 diet supplemented with 50 nM vitamin B12 (*Figure 5D, E*). Taken together, these results suggested that SAM depletion is the primary cause of DTT toxicity.

The genetic and methionine supplementation experiments suggested that SAM depletion could be the primary cause of DTT toxicity. To establish that DTT indeed resulted in the depletion of SAM, we quantified SAM levels in animals with and without exposure to DTT. The exposure of the wild-type N2 animals to DTT resulted in a significant reduction in their SAM levels (*Figure 5F*). To study whether SAM depletion by DTT depended on the gene *rips-1*, we quantified SAM levels in *rips-1(jsn11)* animals. *rips-1(jsn11)* animals did not have significant changes in SAM levels upon exposure to DTT (*Figure 5F*). Although nonsignificant, the SAM levels appeared to be higher in *rips-1(jsn11)* animals upon DTT exposure. It is unclear why DTT exposure resulted in the increased SAM levels in *rips-1(jsn11)* animals. Nevertheless, these results establish that DTT exposure depletes SAM levels in N2, but not *rips-1(jsn11)* animals.

SAM is the universal methyl donor in the cell and regulates a vast array of cellular activities, including the synthesis of phosphatidylcholine by methylation of phosphoethanolamine (*Vance and Vance, 2004*). Several of the SAM functions, including lipogenesis (*Walker et al., 2011*), gene expression changes (*Giese et al., 2020*), innate immunity (*Ding et al., 2015*; *Nair et al., 2022*), are regulated by phosphatidylcholine. Phosphatidylcholine can also be synthesized from choline by an alternate pathway independent of SAM (*Vance and Vance, 2004*). Importantly, the SAM-dependent

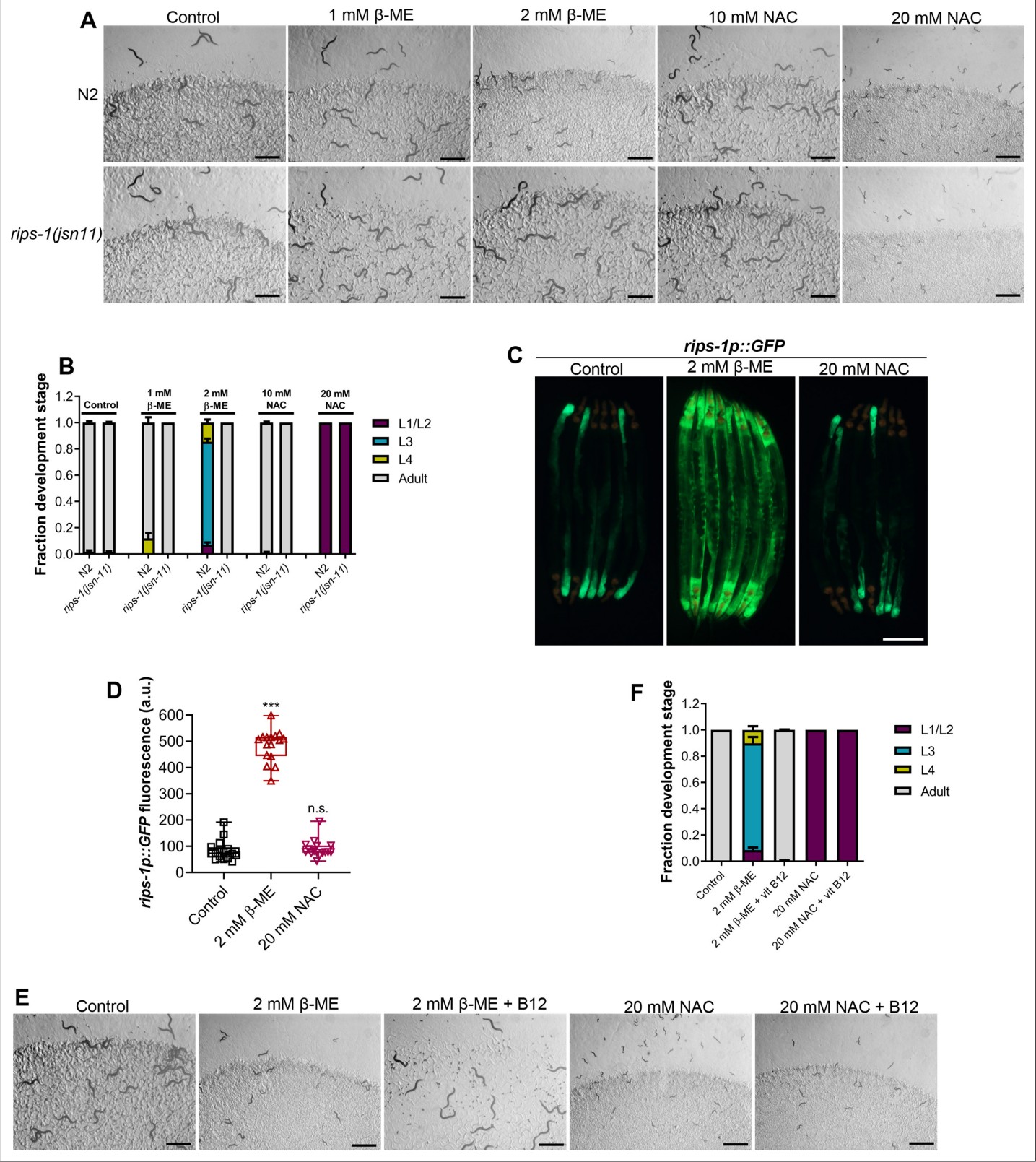

**Figure 4.** β-Mercaptoethanol (β-ME), but not *N*-acetylcysteine (NAC), shares toxicity pathway with dithiothreitol (DTT). (**A**) Representative images of N2 and *rips-1(jsn11)* animals after 72 hr of hatching at 20°C on *E. coli* OP50 diet containing no supplements (control), 1 mM β-ME, 2 mM β-ME, 10 mM NAC, and 20 mM NAC. Scale bar = 1 mm. (**B**) Quantification of different developmental stages of N2 and *rips-1(jsn11)* animals after 72 hr of hatching at 20°C on *E. coli* OP50 diet containing no supplements (control), 1 mM β-ME, 2 mM β-ME, 10 mM NAC, and 20 mM NAC (*n* = 3 biological replicates; animals

*Figure 4 continued on next page*

*Figure 4 continued*

per condition per replicate >50). (**C**) Representative fluorescence (top) and the corresponding brightfield (bottom) images of *rips-1p::GFP* animals grown on *E. coli* OP50 diet without any supplements until the young adult stage, followed by incubation on *E. coli* OP50 diet containing no supplement (control), 2 mM β-ME, or 20 mM NAC for 8 hr. Scale bar = 200 µm. (**D**) Quantification of GFP levels of *rips-1p::GFP* animals grown on *E. coli* OP50 diet without any supplements until the young adult stage, followed by incubation on *E. coli* OP50 diet containing no supplement (control), 2 mM β-ME, or 20 mM NAC for 8 hr. p values are relative to the control. ***p < 0.001 via the *t*-test. n.s., nonsignificant (*n* = 16 worms each). (**E**) Representative images of N2 animals after 72 hr of hatching at 20°C on *E. coli* OP50 diet containing no supplements (control), 2 mM β-ME, 2 mM β-ME supplemented with 50 nM vitamin B12, 20 mM NAC, and 20 mM NAC supplemented with 50 nM vitamin B12. Scale bar = 1 mm. (**F**) Quantification of different developmental stages of N2 animals after 72 hr of hatching at 20°C on *E. coli* OP50 diet containing no supplements (control), 2 mM β-ME, 2 mM β-ME supplemented with 50 nM vitamin B12, 20 mM NAC, and 20 mM NAC supplemented with 50 nM vitamin B12 (*n* = 3 biological replicates; animals per condition per replicate >50).

The online version of this article includes the following source data for figure 4:

**Source data 1.** β-Mercaptoethanol (β-ME), but not *N*-acetylcysteine (NAC), shares toxicity pathway with dithiothreitol (DTT).

phenotypes that require phosphatidylcholine can be rescued by supplementation of choline (***Ding et al., 2015***; ***Giese et al., 2020***; ***Walker et al., 2011***). We asked whether choline could also attenuate DTT toxicity. Supplementation of choline rescued the developmental defects in N2 and *metr-1(ok521)* animals on an *E. coli* OP50 diet containing 10 mM DTT (***Figure 5G, H***). The rescue of DTT toxicity was only partial in *sams-1(ok2946)* animals (***Figure 5G, H***). These results suggested that phosphatidyl-choline is a major SAM product responsible for combating DTT toxicity. However, the partial rescue of DTT toxicity by choline supplementation in *sams-1(ok2946)* animals suggested that other SAM-related functions may also be involved in attenuating DTT toxicity.

## DTT causes ER stress in part by modulation of the methionine–homocysteine cycle

Disturbance in the methionine–homocysteine cycle (accumulation of homocysteine and depletion of SAM) is known to result in ER stress (***Hou et al., 2014***; ***Walker et al., 2011***; ***Werstuck et al., 2001***). Since DTT causes ER stress, we asked what the contribution of methionine–homocysteine cycle perturbation in the DTT-induced ER stress was. To this end, we studied the effect of vitamin B12 and methionine supplementation on DTT-induced upregulation of *hsp-4p::GFP*. Supplementation of vitamin B12 and methionine suppressed DTT-induced upregulation of *hsp-4p::GFP* (***Figure 6A, B***). Choline supplementation is known to suppress the ER stress caused by the depletion of SAM (***Hou et al., 2014***). We observed that supplementation of choline resulted in suppression of DTT-induced upregulation of *hsp-4p::GFP* (***Figure 6A, B***). Thus, these results suggested that DTT causes ER stress in part by modulation of the methionine–homocysteine cycle.

## High levels of DTT cause toxicity via the methionine–homocysteine cycle and ER proteotoxic stress

Next, we explored the role of ER stress in DTT-mediated development retardation. First, we tested whether a lower concentration of DTT, which results in development retardation, would lead to ER stress. Exposure to 5 mM DTT resulted in the upregulation of *hsp-4p::GFP* to an intermediate level between the control and 10 mM DTT exposure levels (***Figure 6—figure supplement 1A, B***), indicating that lower concentrations of DTT also cause ER stress. The ER UPR pathways alleviate enhanced proteotoxic stress, and therefore, the animals having mutations in the UPR pathways have enhanced development retardation on chemical stressors such as tunicamycin (***Shen et al., 2001***). Surprisingly, we observed that in the presence of DTT, the mutants of different UPR pathways showed development retardation similar to the N2 animals (***Figure 6C–G***). Next, we tested whether vitamin B12 could alleviate DTT toxicity in the different UPR mutants. Similar to the wild-type N2 animals, supplementation of 50 nM vitamin B12 resulted in the rescue of both 5 and 10 mM DTT toxicity in *atf-6(ok551)* and *pek-1(ok275)* animals (***Figure 6—figure supplement 1C–F***). On the other hand, while supplementation of 50 nM vitamin B12 rescued the development of *ire-1(v33)* and *xbp-1(tm2482)* animals on 5 mM DTT, it failed to rescue their development on 10 mM DTT (***Figure 6H–J*** and ***Figure 6—figure supplement 2***). Taken together, these results suggested that at lower DTT concentrations, the primary cause of DTT toxicity was the modulation of the methionine–homocysteine cycle, and attenuation of DTT toxicity did not require a functional UPR. On the other hand, at higher DTT concentrations, the toxicity was

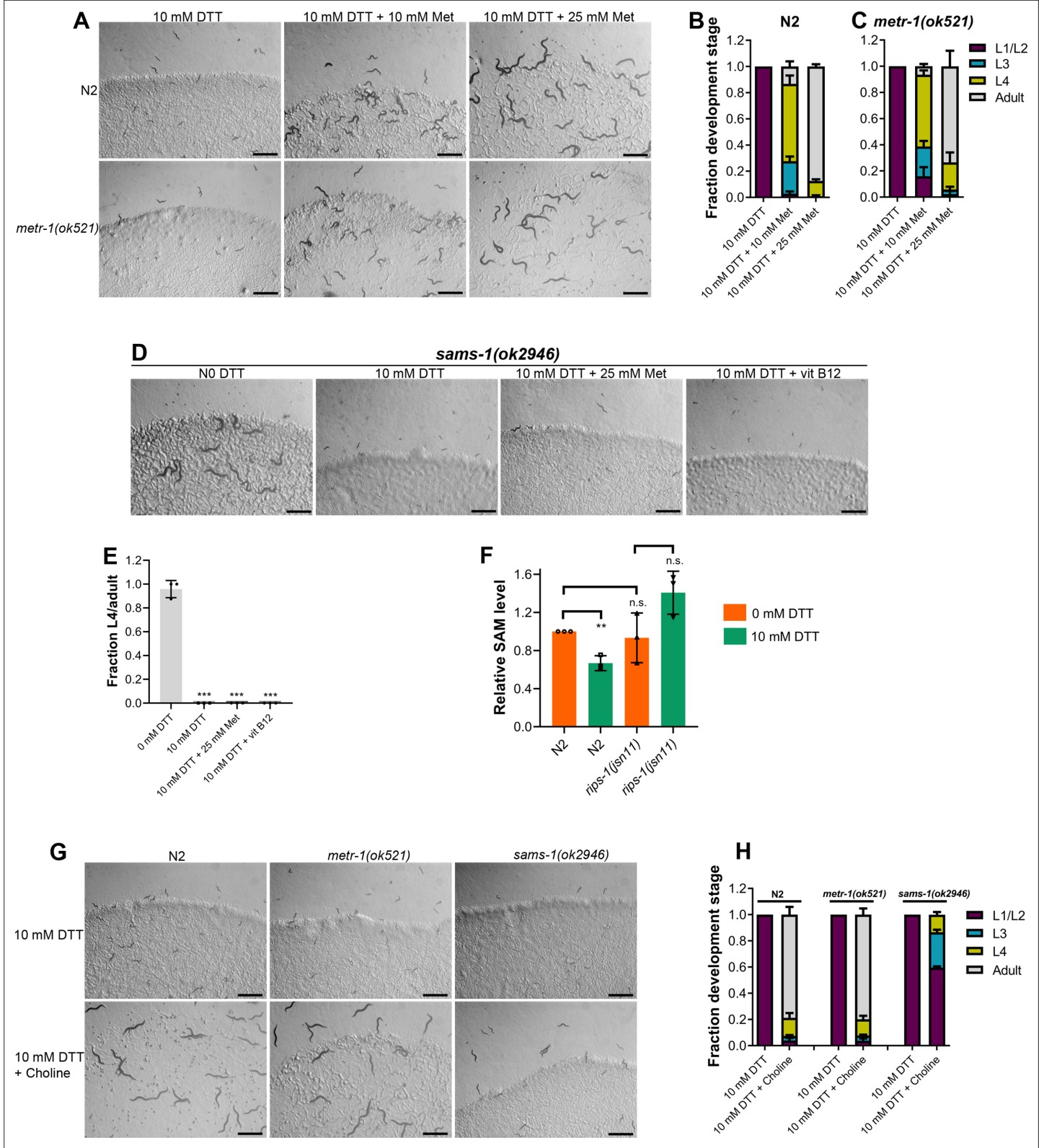

**Figure 5.** Methionine and choline supplementation alleviate dithiothreitol (DTT) toxicity. (**A**) Representative images of N2 and *metr-1(ok521)* animals after 5 days of hatching at 20°C on *E. coli* OP50 diet containing 10 mM DTT, 10 mM DTT supplemented with 10 mM methionine, and 10 mM DTT supplemented with 25 mM methionine. Scale bar = 1 mm. Quantification of different developmental stages of N2 (**B**) and *metr-1(ok521)* (**C**) animals after 5 days of hatching at 20°C on *E. coli* OP50 diet containing 10 mM DTT, 10 mM DTT supplemented with 10 mM methionine, and 10 mM DTT

*Figure 5 continued on next page*

Figure 5 continued

supplemented with 25 mM methionine (*n* = 3 biological replicates; animals per condition per replicate >60). (**D**) Representative images of *sams-1(ok2946)* animals grown on *E. coli* OP50 diet containing 0 mM DTT, 10 mM DTT, 10 mM DTT supplemented with 25 mM methionine, and 10 mM DTT supplemented with 50 nM vitamin B12. The animals were grown for 5 days on the plates containing methionine and for 3 days under all other conditions. Scale bar = 1 mm. (**E**) Fraction L4 or adult *sams-1(ok2946)* animals grown on *E. coli* OP50 diet containing 0, 10, and 10 mM DTT supplemented with 25 mM methionine, and 10 mM DTT supplemented with 50 nM vitamin B12. The animals were grown for 5 days on the plates containing methionine and for 3 days under all other conditions. ***p < 0.001 via the *t*-test (*n* = 3 biological replicates; animals per condition per replicate >50). (**F**) Relative *S*-adenosylmethionine (SAM) levels in N2 and *rips-1(jsn11)* animals grown on *E. coli* OP50 diet containing 0 mM DTT until the young adult stage, followed by incubation on *E. coli* OP50 diet containing 0 or 10 mM DTT for 12 hr. **p < 0.01 via the *t*-test. n.s., nonsignificant (*n* = 3 biological replicates). (**G**) Representative images of N2, *metr-1(ok521)*, and *sams-1(ok2946)* animals after 4 days of hatching at 20°C on *E. coli* OP50 diet containing 10 and 10 mM DTT supplemented with 80 mM choline. Scale bar = 1 mm. (**H**) Quantification of different developmental stages of N2, *metr-1(ok521)*, and *sams-1(ok2946)* animals after 4 days of hatching at 20°C on *E. coli* OP50 diet containing 10 and 10 mM DTT supplemented with 80 mM choline (*n* = 3 biological replicates; animals per condition per replicate >50).

The online version of this article includes the following source data for figure 5:

**Source data 1.** Methionine and choline supplementation alleviate dithiothreitol (DTT) toxicity.

an outcome of both the modulation of the methionine–homocysteine cycle and the ER proteotoxic stress, and attenuation of the DTT toxicity required a functional IRE-1/XBP-1 UPR pathway.

## Discussion

DTT is a potent reducing agent that effectively reduces protein disulfide bonds (*Cleland, 1964*; *Konigsberg, 1972*), and therefore, has a wide range of applications in protein biochemistry, cell biology, and the pharmaceutical industry. Its antioxidant properties also make it an important candidate drug molecule to treat various pathological conditions associated with oxidative stress (*Heidari et al., 2018*; *Heidari et al., 2016a*; *Li et al., 2019*; *Offen et al., 1996*). However, the complete spectrum of physiological effects of DTT inside a cell is not fully understood. The ER is known to be a primary target for DTT inside the cell due to its oxidative nature (*Merksamer et al., 2008*). The oxidative nature of the ER is crucial for the formation of disulfide bonds and the proper folding of proteins. DTT leads to the reduction of disulfide bonds in the ER, consequently leading to protein misfolding and ER stress. DTT has widely been used as an ER-specific stressor (*Braakman et al., 1992*; *Oslowski and Urano, 2011*). Increased ER stress is associated with the activation of cell death programs (*Sano and Reed, 2013*), and therefore, the toxicity of DTT has been associated with increased ER stress (*Li et al., 2011*; *Qin et al., 2010*; *Xiang et al., 2016*). In addition to the ER stress, DTT is shown to exert its toxic effects by paradoxically increasing ROS production (*Held et al., 1996*; *Held and Melder, 1987*). In the current study, we showed that in *C. elegans*, DTT causes toxicity by modulating the methionine–homocysteine cycle. By upregulating the expression of SAM-dependent methyltransferase gene *rips-1*, DTT leads to the depletion of SAM that results in toxicity. Supplementation of vitamin B12 and methionine rescues DTT toxicity by repleting SAM levels (*Figure 7*). Therefore, our study indicates that the effects of DTT are not limited to the ER. Indeed, our data showed that DTT also upregulates the mitochondrial UPR via the methionine–homocysteine cycle and argues against DTT being an ER-specific stressor.

Our genetic screen for DTT-resistant mutants resulted in the isolation of 12 loss-of-function alleles of the same gene, the SAM-dependent methyltransferase *rips-1*. Mutation in no other gene, including the *rips-1* related methyltransferases, which have a comparable size to *rips-1*, was recovered, suggesting that the screen had reached a saturation level. These results indicated that DTT toxicity could be very specifically mediated via the modulation of the methionine–homocysteine cycle. Indeed, we observed that DTT toxicity at 5 mM concentration could be rescued in all of the different ER UPR pathway mutants by supplementation of vitamin B12 (*Figure 6*, *Figure 6—figure supplement 1*, and *Figure 6—figure supplement 2*). These results suggested that at 5 mM concentration, the toxic effects of DTT were not due to the ER proteotoxic stress and primarily were due to the modulation of the methionine–homocysteine cycle. However, DTT toxicity at 10 mM DTT concentration could not be rescued by vitamin B12 supplementation in *ire-1(v33)* and *xbp-1(tm2482)* animals, while it was rescued in the wild-type as well as *atf-6(ok551)* and *pek-1(ok275)* animals. This indicated that at 10 mM concentration, DTT toxicity occurred both by the modulation of the methionine–homocysteine

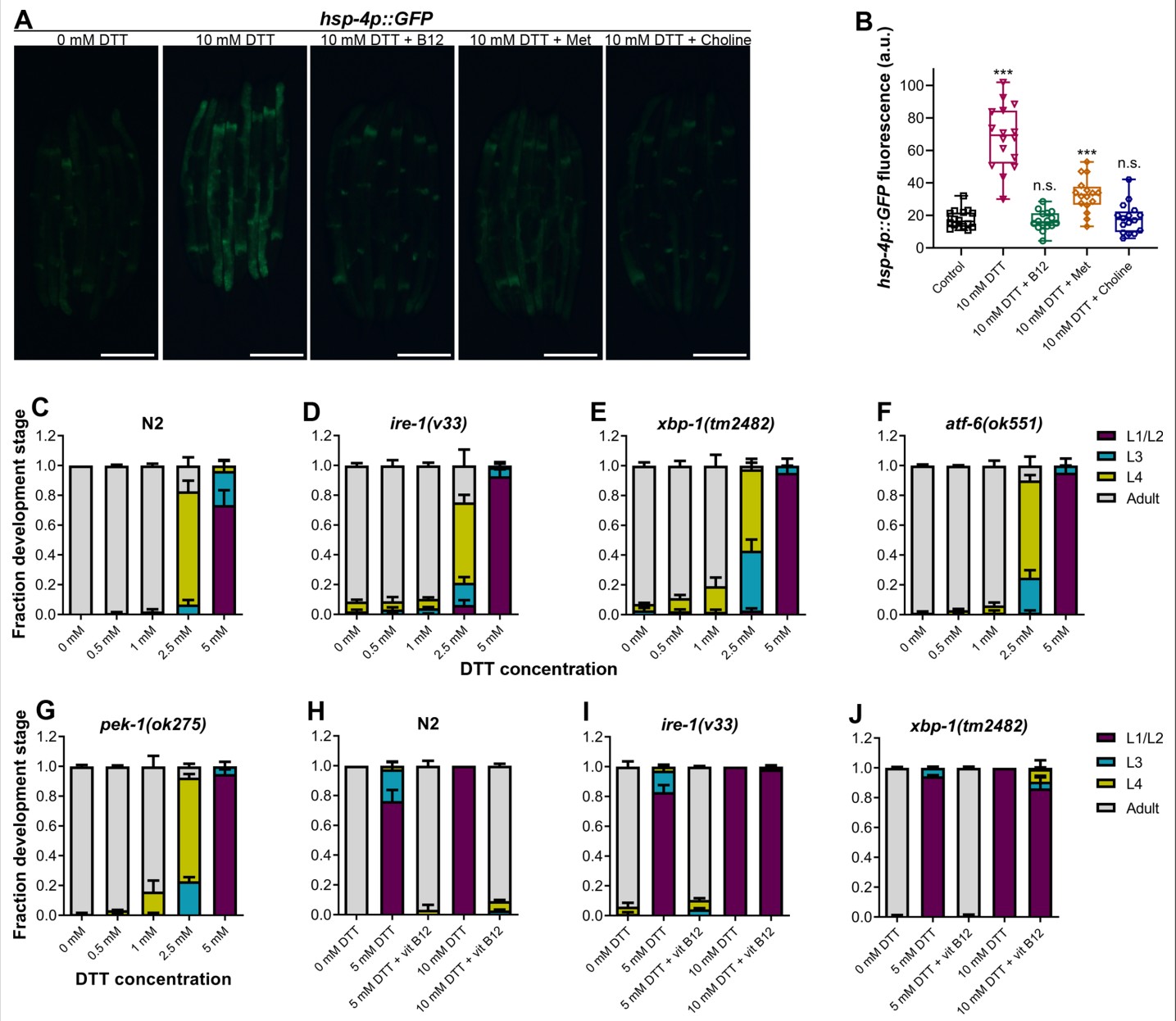

**Figure 6.** High levels of dithiothreitol (DTT) cause toxicity via the methionine–homocysteine cycle and ER proteotoxic stress. (**A**) Representative fluorescence images of *hsp-4p::GFP* animals grown on *E. coli* OP50 diet containing 0 mM DTT until the young adult stage, followed by incubation on *E. coli* OP50 diet containing 0, 10, and 10 mM DTT supplemented with 50 nM vitamin B12, 10 mM DTT supplemented with 25 mM methionine, and 10 mM DTT supplemented with 80 mM choline for 24 hr. Scale bar = 200 µm. (**B**) Quantification of GFP levels of *hsp-4p::GFP* animals grown on *E. coli* OP50 diet containing 0 mM DTT until the young adult stage, followed by incubation on *E. coli* OP50 diet containing 0, 10, 10 mM DTT supplemented with 50 nM vitamin B12, 10 mM DTT supplemented with 25 mM methionine, and 10 mM DTT supplemented with 80 mM choline for 24 hr. ***$p < 0.001$ via the *t*-test. n.s., nonsignificant (*n* = 16 worms each). Quantification of different developmental stages of N2 (**C**), *ire-1(v33)* (**D**), *xbp-1(tm2482)* (**E**), *atf-6(ok551)* (**F**), and *pek-1(ok275)* (**G**) animals on various concentrations of DTT on *E. coli* OP50 diet. The *ire-1(v33)* animals were grown for 84 hr while all other animals were grown for 72 hr at 20°C (*n* = 3 biological replicates; animals per condition per replicate >80). Quantification of development of N2 (**H**), *ire-1(v33)* (**I**), and *xbp-1(tm2482)* (**J**) animals on *E. coli* OP50 plates containing 0, 5, and 5 mM DTT supplemented with 50 nM vitamin B12, 10 mM DTT, and 10 mM DTT supplemented with 50 nM vitamin B12 (*n* = 3 biological replicates; animals per condition per replicate >80).

The online version of this article includes the following source data and figure supplement(s) for figure 6:

**Source data 1.** High levels of dithiothreitol (DTT) cause toxicity via the methionine–homocysteine cycle and ER proteotoxic stress.

**Figure supplement 1.** The ATF-6 and PEK-1 pathways are not involved in combating dithiothreitol (DTT) toxicity.

**Figure supplement 1—source data 1.** The ATF-6 and PEK-1 pathways are not involved in combating dithiothreitol (DTT) toxicity.

*Figure 6 continued on next page*

*Figure 6 continued*

**Figure supplement 2.** Counteracting high, but not low, levels of dithiothreitol (DTT) requires a functional IRE-1/XBP-1 unfolded protein response (UPR) pathway.

cycle and ER proteotoxic stress, and the animals needed a functional IRE-1/XBP-1 pathway to counteract it.

One question that arises from the current study concerns the substrate for the SAM-methyltransferase RIPS-1. Because DTT results in the induction of *rips-1* expression, one possibility is that DTT is the substrate for RIPS-1, and *rips-1* expression is induced in response to DTT as a defense mechanism. This hypothesis is supported by studies that showed that multiple thiol reagents, including DTT and β-ME, could be methylated by SAM and SAM-dependent methyltransferases (**Bremer and Greenberg, 1961**; **Coiner et al., 2006**; **Maldonato et al., 2021**). The methylation of external thiols could be a defense response for some organisms. For example, the dithiol metabolite gliotoxin produced by *Aspergillus fumigatus* inhibits the growth of other fungi, including *A. niger*, and triggers a defense response in *A. niger* resulting in upregulation of SAM-dependent methyltransferases (**Dolan et al., 2014**; **Manzanares-Miralles et al., 2016**; **Owens et al., 2015**). Methylation of gliotoxin by SAM-dependent methyltransferases confers *A. niger* protection against the dithiol metabolite. Incidentally, the same methyltransferases that are required for the defense mechanism against gliotoxin are induced by DTT (**MacKenzie et al., 2005**; **Manzanares-Miralles et al., 2016**). Therefore, upregulation of methyltransferases might be a general defense response against multiple thiol reagents.

Methionine metabolism and methyltransferases have important roles in regulating health and lifespan (**Parkhitko et al., 2019**), and defects in the methionine–homocysteine cycle have been linked with a large number of pathological conditions (**Finkelstein, 2006**; **Li et al., 2020**; **Sanderson et al., 2019**). Therefore, the methionine–homocysteine cycle has become an important intervention point for improving lifespan and health (**Konno et al., 2017**; **Lam et al., 2021**; **Li et al., 2017**; **Parkhitko et al., 2019**; **Sanderson et al., 2019**). In the current study, we discovered that DTT modulates the methionine–homocysteine cycle resulting in low SAM levels. Reduction of SAM levels is known to enhance the lifespan and metabolic health of a large variety of organisms (**Parkhitko et al., 2019**). Therefore, it will be intriguing to test whether, at nontoxic concentrations, DTT could have health-improving effects via the methionine–homocysteine cycle.

In the future, it will be interesting to study whether the modulation of the methionine–homocysteine cycle by DTT is a conserved feature across different organisms. In the fungus *Aspergillus niger*, DTT is known to strongly upregulate the expression of two SAM-dependent methyltransferases (**MacKenzie et al., 2005**). Therefore, it is likely possible that DTT modulates the methionine–homocysteine cycle in *Aspergillus niger*. Similarly, DTT results in increased levels of cysteine in Chinese hamster V79 cells

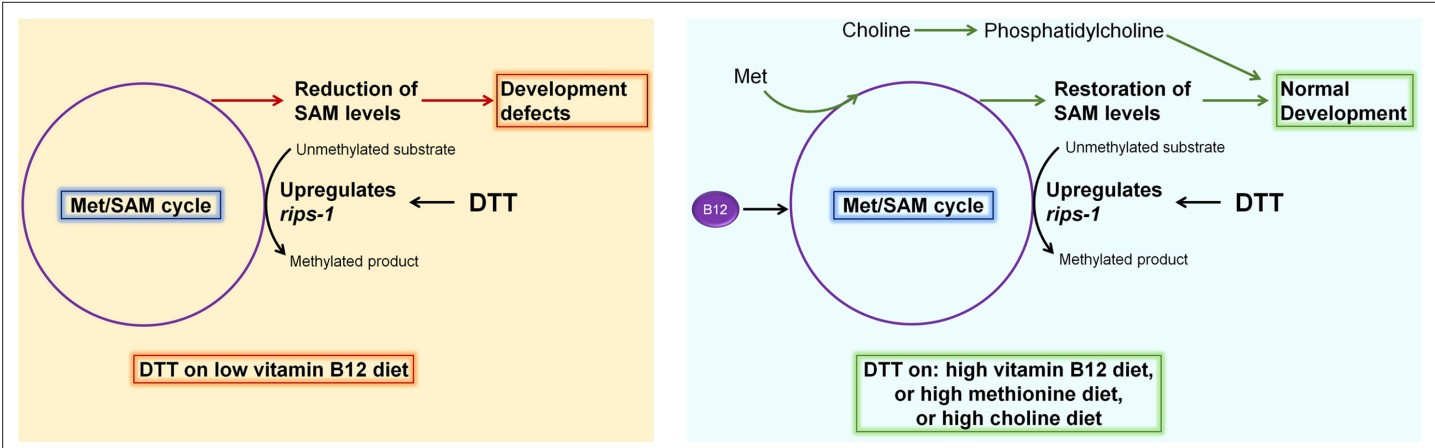

**Figure 7.** Model for dithiothreitol (DTT) toxicity via the methionine–homocysteine cycle. DTT causes upregulation of the *S*-adenosylmethionine (SAM)-dependent methyltransferase gene *rips-1*. On low vitamin B12 diet, DTT leads to the depletion of SAM that results in toxicity. Supplementation of vitamin B12 and methionine attenuates DTT toxicity by restoring SAM levels. Supplementation of choline also rescues DTT toxicity by repleting phosphatidylcholine via an SAM-independent pathway.

(*Dennis et al., 1989*). Cysteine is synthesized from homocysteine by the transsulfuration pathway, and increased amounts of homocysteine are known to increase cysteine levels (*Sbodio et al., 2019*). Therefore, it is possible that DTT modulates the methionine–homocysteine cycle in Chinese hamster V79 cells resulting in increased levels of homocysteine and, consequently, increased cysteine levels. Overall, these studies suggest that modulation of the methionine–homocysteine cycle by DTT could be a conserved feature across different organisms. Future research deciphering the mechanism and relevance of activation of SAM-dependent methyltransferases by thiol reagents DTT and β-ME will aid in better understanding the physiological effects of these, and possibly, other thiol reagents.

# Materials and methods

## Key resources table

| Reagent type (species) or resource | Designation | Source or reference | Identifiers | Additional information |
|---|---|---|---|---|
| Strain, strain background (*Escherichia coli*) | OP50 | *Caenorhabditis* Genetics Center (CGC) | OP50 | |
| Strain, strain background (*E. coli*) | HT115 | Source BioScience | HT115 | |
| Strain, strain background (*P. aeruginosa*) | PA14 *gacA* | Meta Kuehn laboratory | PA14 *gacA* | |
| Strain, strain background (*Comamonas aquatica*) | *C. aquatica* DA1877 | CGC | DA1877 | |
| Strain, strain background (*Serratia marcescens*) | *S. marcescens* Db11 | CGC | Db11 | |
| Strain, strain background (*Salmonella enterica* serovar Typhimurium 1344) | *S. enterica* | Frederick M. Ausubel laboratory | Typhimurium 1344 | |
| Strain, strain background (*Caenorhabditis elegans*) | N2 Bristol | CGC | N2 | |
| Strain, strain background (*C. elegans*) | wwIs24 [acdh-1p::GFP + unc-119(+)] | CGC | VL749 | |
| Strain, strain background (*C. elegans*) | metr-1(ok521) | CGC | RB755 | |
| Strain, strain background (*C. elegans*) | mmcm-1(ok1637) | CGC | RB1434 | |
| Strain, strain background (*C. elegans*) | sams-1(ok2946) | CGC | VC2428 | |
| Strain, strain background (*C. elegans*) | ire-1(v33) | CGC | RE666 | |
| Strain, strain background (*C. elegans*) | atf-6(ok551) | CGC | RB772 | |
| Strain, strain background (*C. elegans*) | pek-1(ok275) | CGC | RB545 | |
| Strain, strain background (*C. elegans*) | zcIs13 [hsp-6p::GFP + lin-15(+)] | CGC | SJ4100 | |
| Strain, strain background (*C. elegans*) | zcIs4 [hsp-4::GFP] | CGC | SJ4005 | |
| Strain, strain background (*C. elegans*) | rips-1(gk231506) | CGC | VC20288 | |
| Strain, strain background (*C. elegans*) | xbp-1(tm2482) | National Bioresource Project, Japan | xbp-1(tm2482) | |
| Strain, strain background (*C. elegans*) | rips-1(jsn1) | This study | JSJ1 | Materials and methods |
| Strain, strain background (*C. elegans*) | rips-1(jsn2) | This study | JSJ2 | Materials and methods |

*Continued on next page*

*Continued*

| Reagent type (species) or resource | Designation | Source or reference | Identifiers | Additional information |
|---|---|---|---|---|
| Strain, strain background (*C. elegans*) | *rips-1(jsn3)* | This study | JSJ3 | Materials and methods |
| Strain, strain background (*C. elegans*) | *rips-1(jsn4)* | This study | JSJ4 | Materials and methods |
| Strain, strain background (*C. elegans*) | *rips-1(jsn5)* | This study | JSJ5 | Materials and methods |
| Strain, strain background (*C. elegans*) | *rips-1(jsn6)* | This study | JSJ6 | Materials and methods |
| Strain, strain background (*C. elegans*) | *rips-1(jsn7)* | This study | JSJ7 | Materials and methods |
| Strain, strain background (*C. elegans*) | *rips-1(jsn8)* | This study | JSJ8 | Materials and methods |
| Strain, strain background (*C. elegans*) | *rips-1(jsn9)* | This study | JSJ9 | Materials and methods |
| Strain, strain background (*C. elegans*) | *rips-1(jsn10)* | This study | JSJ10 | Materials and methods |
| Strain, strain background (*C. elegans*) | *rips-1(jsn11)* | This study | JSJ11 | Materials and methods |
| Strain, strain background (*C. elegans*) | *rips-1(jsn12)* | This study | JSJ12 | Materials and methods |
| Strain, strain background (*C. elegans*) | *jsnIs1[rips-1p::GFP + myo-2p::mCherry]* | This study | JSJ13 | Materials and methods |
| Strain, strain background (*C. elegans*) | *rips-1(jsn11);jsnEx1 [rips-1p::rips-1::SL2::GFP + myo-2p::mCherry]* | This study | | Materials and methods |
| Strain, strain background (*C. elegans*) | *rips-1(jsn11);jsnEx2 [rips-1p::rips-1::SL2::GFP + myo-2p::mCherry]* | This study | | Materials and methods |
| Strain, strain background (*C. elegans*) | *jsnEx1 [rips-1p::rips-1::SL2::GFP + myo-2p::mCherry]* | This study | | Materials and methods |
| Recombinant DNA reagent | *rips-1p::rips-1::SL2::GFP* (plasmid) | This study | | Materials and methods |
| Recombinant DNA reagent | *rips-1p::GFP (plasmid)* | This study | | Materials and methods |
| Sequence-based reagent | Pan-act_qPCR_F | This study | qPCR primers | TCGGTATGGGACAGAAGGAC |
| Sequence-based reagent | Pan-act_qPCR_R | This study | qPCR primers | CATCCCAGTTGGTGACGATA |
| Sequence-based reagent | R08E5.1_qPCR_F | This study | qPCR primers | CAATGACAGGGCCAACATGG |
| Sequence-based reagent | R08E5.1_qPCR_R | This study | qPCR primers | GCAGTATACAACACATTTAGGGGA |
| Sequence-based reagent | rips-1_qPCR_F | This study | qPCR primers | ACAACACGTGGACAACGGTAT |
| Sequence-based reagent | rips-1_qPCR_R | This study | qPCR primers | TTGCTGACGGCGAGGTTAAA |
| Sequence-based reagent | R08F11.4_qPCR_F | This study | qPCR primers | AGACCTACGGGAAGATGGCT |
| Sequence-based reagent | R08F11.4_qPCR_R | This study | qPCR primers | CGCAACATAGTGCATCTGGC |
| Sequence-based reagent | K12D9.1_qPCR_F | This study | qPCR primers | CCGATGGATCCGACTTCCAG |
| Sequence-based reagent | K12D9.1_qPCR_R | This study | qPCR primers | TCGAAGCAACCAGTCCAGTC |
| Sequence-based reagent | rips-1_promoter_F | This study | PCR primers (Cloning) | AAGGTCGACCATTGCTTACTGCTAGGTTCT |
| Sequence-based reagent | rips-1_promoter_R | This study | PCR primers (Cloning) | AGCGGATCCAGTGATCAATTGAACATACAC |

*Continued on next page*

*Continued*

| Reagent type (species) or resource | Designation | Source or reference | Identifiers | Additional information |
|---|---|---|---|---|
| Sequence-based reagent | rips-1_gene_R | This study | PCR primers (Cloning) | AGTGGATCCCTAATTTTTCTGGGCACAATAC |
| Software, algorithm | GraphPad Prism 8 | GraphPad Software | RRID:SCR_002798 | https://www.graphpad.com/scientificsoftware/prism/ |
| Software, algorithm | Photoshop CS5 | Adobe | RRID:SCR_014199 | https://www.adobe.com/products/photoshop.html |
| Software, algorithm | ImageJ | NIH | RRID:SCR_003070 | https://imagej.nih.gov/ij/ |

## Bacterial strains

The following bacterial strains were used: *E. coli* OP50, *E. coli* HT115(DE3), *P. aeruginosa* PA14 *gacA* mutant, *C. aquatica* DA1877, *S. marcescens* Db11, and *S. enterica* serovar Typhimurium 1344. The cultures of these bacteria were grown in Luria–Bertani (LB) broth at 37°C. The nematode growth medium (NGM) plates were seeded with the different bacterial cultures and incubated at room temperature for at least 2 days before using for experiments.

## *C. elegans* strains and growth conditions

*C. elegans* hermaphrodites were maintained at 20°C on NGM plates seeded with *E. coli* OP50 as the food source (Brenner, 1974) unless otherwise indicated. Bristol N2 was used as the wild-type control unless otherwise indicated. The following strains were used in the study: VL749 *wwIs24 [acdh-1p::GFP + unc-119(+)]*, RB755 *metr-1(ok521)*, RB1434 *mmcm-1(ok1637)*, VC2428 *sams-1(ok2946)*, RE666 *ire-1(v33)*, *xbp-1(tm2482)*, RB772 *atf-6(ok551)*, RB545 *pek-1(ok275)*, SJ4100 *zcIs13 [hsp-6p::GFP + lin-15(+)]*, SJ4005 *zcIs4 [hsp-4::GFP]*, and VC20288 *rips-1(gk231506)*. Some of the strains were obtained from the *Caenorhabditis* Genetics Center (University of Minnesota, Minneapolis, MN). The *xbp-1(tm2482)* animals were obtained from National Bioresource Project, Japan.

## Preparation of NGM plates with different supplements

The following supplements with their product numbers were obtained from HiMedia BioSciences: DTT (#RM525), β-mercaptoethanol (#MB041), NAC (#RM3142), vitamin B12 (cyanocobalamin) (#PCT0204), methionine (#PCT0315), and choline chloride (#TC102). All of the supplements that were added to the NGM plates were dissolved in water to prepare a stock solution. The supplements were added to the nematode growth media just before pouring into plates to obtain a desired final concentration.

## *C. elegans* development assays

Synchronized *C. elegans* eggs were obtained by transferring 15–20 gravid adult hermaphrodites on NGM plates for egg-laying for 2 hr. After 2 hr, the gravid adults were removed. The synchronized eggs were incubated at 20°C for 72 hr. After that, the animals in different development stages (L1/L2, L3, L4, and adult) were quantified. Since *ire-1(v33)* animals develop slightly slower than wild-type animals, the development stages in *ire-1(v33)* animals were quantified after 84 hr of incubation at 20°C. For choline and methionine supplementation assays, the synchronized eggs were incubated at 20°C for 4 and 5 days, respectively, before quantifying the development stages. Representative images of the NGM plates at the time of quantification of development were also captured. At least three independent experiments (biological replicates) were performed for each condition.

## RNA interference

RNAi was used to generate loss-of-function RNAi phenotypes by feeding nematodes *E. coli* strain HT115(DE3) expressing double-stranded RNA homologous to a target gene. *E. coli* with the appropriate vectors were grown in LB broth containing ampicillin (100 µg/ml) at 37°C overnight and plated onto NGM plates containing 100 µg/ml ampicillin and 3 mM isopropyl β-D-thiogalactoside (RNAi plates). The RNAi-expressing bacteria were allowed to grow overnight at 37°C on RNAi plates without DTT. On the other hand, the RNAi plates with 10 mM DTT were kept at room temperature for 2 days upon seeding with bacterial cultures. The wild-type N2 worms were first synchronized on RNAi plates without DTT and allowed to develop at 20°C for 3 days. The gravid adults obtained from RNAi plates without DTT were transferred to corresponding RNAi plates containing 10 mM DTT and allowed to

lay eggs for 2 hr. The gravid adults were removed, and the eggs were allowed to develop at 20°C for 72 hr before quantifying the development stages. The RNAi clones were from the Ahringer RNAi library and were verified by sequencing.

## Rescue of DTT-mediated development retardation by vitamin B12 supplementation

To study whether the DTT-mediated development retardation could be reversed by supplementation of vitamin B12, 2 µl of 750 µM vitamin B12 (total 1.5 nmol) was added to the center of a plate with L1-arrested animals after 72 hr of hatching on *E. coli* OP50 with 10 mM DTT. To the control plate with L1-arrested animals after 72 hr of hatching on *E. coli* OP50 with 10 mM DTT, 2 µl of water was added to the center. After that, the plates were incubated at 20°C for 48 hr, followed by the quantification of development stages.

## Forward genetic screen for DTT-resistant mutants

An EMS mutagenesis selection screen (*Singh, 2021*) was performed using the wild-type N2 strain. Approximately 2500 synchronized late L4 larvae were treated with 50 mM EMS for 4 hr and then washed three times with M9 medium. The washed animals (P0 generation) were then transferred to 9 cm NGM plates containing *E. coli* OP50 and allowed to lay eggs (F1 progeny) overnight. The P0s were then washed away with M9 medium, while the F1 eggs remained attached to the bacterial lawn. The F1 eggs were allowed to grow to adulthood. The adult F1 animals were bleached to obtain F2 eggs. The F2 eggs were transferred to *E. coli* OP50 plates containing 10 mM DTT and incubated at 20°C for 72 hr. After that, the plates were screened for animals that developed to L4 or adult stages. Approximately 50,000 haploid genomes were screened, and 12 mutants were isolated. All of the mutants were backcrossed six times with the parental N2 strain before analysis.

## Whole-genome sequencing and data analysis

The genomic DNA was isolated as described earlier (*Singh and Aballay, 2017*). Briefly, the mutant animals were grown at 20°C on NGM plates seeded with *E. coli* OP50 until starvation. The animals were rinsed off the plates with M9, washed three times, incubated in M9 with rotation for 2 hr to eliminate food from the intestine, and washed three times again with M9. Genomic DNA extraction was performed using the Gentra Puregene Kit (Qiagen, Netherlands). DNA libraries were prepared according to a standard Illumina (San Diego, CA) protocol. The DNA was subjected to whole-genome sequencing (WGS) on an Illumina HiSeq sequencing platform using 150 paired-end nucleotide reads. Library preparation and WGS were performed at Clevergene Biocorp Pvt. Ltd, Bengaluru, India.

The whole-genome sequence data were analyzed using the web platform Galaxy. The forward and reverse FASTQ reads, *C. elegans* reference genome Fasta file (ce11M.fa), and the gene annotation file (SnpEff4.3 WBcel235.86) were input into the Galaxy workflow. The low-quality ends of the FASTQ reads were trimmed using the Sickle tool. The trimmed FASTQ reads were mapped to the reference genome Fasta files with the BWA-MEM tool. Using the MarkDuplicates tool, any duplicate reads (mapped to multiple sites) were filtered. Subsequently, the variants were detected using the Free-Bayes tool that finds small polymorphisms, including SNPs, insertions and deletions (indels), multinucleotide polymorphisms, and complex events (composite insertion and substitution events) smaller than the length of a short-read sequencing alignment. The common variants among different mutants were subtracted. The SnpEff4.3 Wbcel235.86 gene annotation file was used to annotate and predict the effects of genetic variants (such as amino acid changes). Finally, the linkage maps for each mutant were generated using the obtained variation.

## Plasmid constructs and generation of transgenic *C. elegans*

For rescue of the DTT-resistance phenotype of *rips-1(jsn11)* animals, the *rips-1* gene along with its promoter region (918 bp upstream) was amplified from genomic DNA of N2 animals. The gene, including its stop codon, was cloned in the pPD95.77 plasmid containing the SL2 sequence before GFP. For generating the *rips-1p::GFP* reporter strain, the promoter region of the *rips-1* gene (918 bp upstream) was amplified from genomic DNA of N2 animals and cloned in the pPD95.77 plasmid before GFP. *rips-1(jsn11)* animals were microinjected with *rips-1p::rips-1::SL2::GFP* plasmid along with pCFJ90 (*myo-2p::mCherry*) as a coinjection marker to generate rescue strains,

*rips-1(jsn11);jsnEx1[rips-1p::rips-1::SL2::GFP + myo-2p::mCherry]* and *rips-1(jsn11);jsnEx2[rips-1p::rips-1::SL2::GFP + myo-2p::mCherry]*. To obtain *rips-1* overexpressing strain, the *rips-1(jsn11)* allele was outcrossed to wild-type N2 animals to obtain *jsnEx1[rips-1p::rips-1::SL2::GFP + myo-2p::mCherry]* animals. N2 wild-type animals were microinjected with *rips-1p::GFP* plasmid along with pCFJ90 as a coinjection marker to generate *rips-1p::GFP* reporter strain. The *rips-1p::rips-1::SL2::GFP* and *rips-1p::GFP* plasmids were used at a concentration of 50 ng/µl, while the coinjection marker was used at a concentration of 5 ng/µl. The plasmids were maintained as extrachromosomal arrays, and at least two independent lines were maintained for each plasmid. The *rips-1p::GFP* array was integrated using UV irradiation to obtain *jsnIs1[rips-1p::GFP + myo-2p::mCherry]* followed by six times back-crossing with the wild-type N2 animals.

## RNA isolation and qRT-PCR

Animals were synchronized by egg laying. Approximately 35 N2 gravid adult animals were transferred to 9 cm *E. coli* OP50 plates without DTT and allowed to lay eggs for 4 hr. The gravid adults were then removed, and the eggs were allowed to develop at 20°C for 72 hr. Subsequently, the synchronized adult animals were collected with M9, washed twice with M9, and then transferred to 9 cm *E. coli* OP50 plates containing 10 mM DTT. The control animals were maintained on *E. coli* OP50 plates without DTT. After the transfer of the animals, the plates were incubated at 20°C for 4 hr. Subsequently, the animals were collected, washed with M9 buffer, and frozen in TRIzol reagent (Life Technologies, Carlsbad, CA). Total RNA was extracted using the RNeasy Plus Universal Kit (Qiagen, Netherlands). A total of 1 µg of total RNA was reverse-transcribed with oligo dT primers using the Transcriptor High Fidelity cDNA Synthesis Kit (Roche) according to the manufacturer's protocols. qRT-PCR was conducted using TB Green fluorescence (TaKaRa) on a LightCycler 480 II System (Roche Diagnostics) in 96-well-plate format. Twenty microliter reactions were analyzed as outlined by the manufacturer (TaKaRa). The relative fold-changes of the transcripts were calculated using the comparative $CT(2^{-\Delta\Delta CT})$ method and normalized to pan-actin (*act-1*, *-3*, *-4*) as described earlier (*Singh and Aballay, 2019a*). All samples were run in triplicate.

## Quantification of SAM levels

Animals were synchronized by egg laying. Approximately 40 N2 and *rips-1(jsn11)* gravid adult animals each were transferred to 9 cm *E. coli* OP50 plates without DTT and allowed to lay eggs for 4–5 hr. For every experiment, synchronization on six 9 cm *E. coli* OP50 plates (three plates for DTT treatment and control each) was carried out for each strain. The gravid adults were then removed, and the eggs were allowed to develop at 20°C for 72 hr. The animals were then collected, washed with M9 buffer, and transferred to 9 cm *E. coli* OP50 plates containing 10 mM DTT and incubated at 20 C for 12 hr. The control animals were maintained on 9 cm *E. coli* OP50 plates without DTT. After that, the animals were collected and washed with M9 buffer three to four times. The semisoft pellet of the animals was weighed (50–70 mg of worm pellet) and stored at −80°C until further processing. The samples were processed for metabolite extraction as described earlier (*Nair et al., 2022*) with appropriate modifications. Briefly, the worm pellets were thawed from −80°C and mixed with an ice-chilled acetonitrile–methanol–water (3:5:2) mixture containing 0.1% formic acid, followed by eight cycles of sonication with 3 s on and 30 s off at an amplitude of 40%. The suspension was centrifuged at 15,000 × *g* at 4°C for 10 min. Then, the supernatant was transferred to a fresh tube and was vacuum dried and stored at −80°C until processed for mass spectrometry.

For mass spectrometry, the sample was reconstituted in ice-chilled water with 0.1% formic acid. The volume of water used to reconstitute the samples depended on the weight of the preprocessed semisoft pellet of the animals, such that 1 µl of water was added per mg of worm pellet. The reconstituted mixture was centrifuged at 15,000 × *g* for 2 min, and 5 µl was injected for LC–MS analysis. The separation of metabolites was achieved on an Acquity UPLC BEH130 C18 column (2.1 × 100 mm, 1.7 µm particles, Waters) at room temperature. The two mobile phases consisted of (A) water with 0.1% formic acid and (B) acetonitrile with 0.1% formic acid. Linear gradients from 100% A to 90% A for 6 min were used for separation at a flow rate of 0.2 ml/min. The effluent was then introduced into the electrospray ionization source of the Synapt G2-Si HD mass spectrometer (Waters). The acquisitions were carried out in the positive ion mode. SAM chloride dihydrochloride was purchased from

Sigma-Aldrich (#A7007) and was used as a standard. SAM levels were quantified by using the peak intensity at 399.1 *m/z* upon integration of SAM chromatogram peak.

### *rips-1p::GFP* induction by DTT

The *rips-1p::GFP* animals were synchronized by egg laying on *E. coli* OP50 plates and incubated at 20°C for 72 hr. Subsequently, the animals were transferred to *E. coli* OP50 plates containing 10 mM DTT, 2 mM β-ME, and 20 mM NAC and incubated at 20°C for 8 hr. The control animals were maintained on *E. coli* OP50 plates without the chemical supplements. After that, the animals were prepared for fluorescence imaging.

### *acdh-1p::GFP* assays

The *acdh-1p::GFP* animals were synchronized by egg laying on different bacterial diets without DTT. The animals were incubated at 20°C for 72 hr. After that, the animals were prepared for fluorescence imaging.

### *hsp-6p::GFP* induction assays

The *hsp-6p::GFP* animals were synchronized by egg laying on *E. coli* OP50 plates without DTT and incubated at 20°C for 72 hr. Subsequently, the animals were transferred to *E. coli* OP50 plates containing 0, 10, and 10 mM DTT supplemented with 50 nM vitamin B12 followed by incubation at 20°C for 24 hr. After that, the animals were prepared for fluorescence imaging.

### *hsp-4p::GFP* induction assays

The *hsp-4p::GFP* animals were synchronized by egg laying on *E. coli* OP50 plates without DTT and incubated at 20°C for 72 hr. Subsequently, the animals were transferred to *E. coli* OP50 plates containing 0, 5, 10, 10 mM DTT supplemented with 50 nM vitamin B12, 10 mM DTT supplemented with 25 mM methionine, and 10 mM DTT supplemented with 80 mM choline chloride followed by incubation at 20°C for 24 hr. After that, the animals were prepared for fluorescence imaging.

## Fluorescence imaging and quantification

Fluorescence imaging was carried out as described previously (*Singh and Aballay, 2019b*). Briefly, the animals were anesthetized using an M9 salt solution containing 50 mM sodium azide and mounted onto 2% agarose pads. The animals were then visualized using a Nikon SMZ-1000 fluorescence stereomicroscope. The fluorescence intensity was quantified using Image J software.

## Quantification and statistical analysis

The statistical analysis was performed with Prism 8 (GraphPad). All error bars represent the standard deviation (SD). The two-sample *t*-test was used when needed, and the data were judged to be statistically significant when $p < 0.05$. In the figures, asterisks (*) denote statistical significance as follows: *$p < 0.05$, **$p < 0.01$, ***$p < 0.001$, as compared with the appropriate controls.

## Acknowledgements

Some strains used in this study were provided by the *Caenorhabditis* Genetics Center (CGC), which is funded by the NIH Office of Research Infrastructure Programs (P40 OD010440). We thank Dr. Pratima Pandey (IISER Mohali) for help with qRT-PCR experiments, Ravi (IISER Mohali) for some technical assistance, Dr. Lolitika Mandal's laboratory for *acdh-1p::GFP* fluorescence imaging, Dr. Samrat Mukhopadhyay's laboratory for the use of sonicator, and HRMS Central Facility (IISER Mohali) for the mass spectrometer.

## Additional information

### Funding

| Funder | Grant reference number | Author |
|---|---|---|
| Indian Institute of Science Education and Research Bhopal | INST/BIO/2019091 | Jogender Singh |
| Indian Institute of Science Education and Research Mohali | Intramural funds | Jogender Singh |
| Science and Engineering Research Board | Startup Research Grant | Jogender Singh |
| Department of Biotechnology, Ministry of Science and Technology, India | Ramalingaswami Re-entry Fellowship | Jogender Singh |
| Department of Science and Technology, Ministry of Science and Technology, India | INSPIRE-SHE Scholarship | Gokul G |
| Department of Biotechnology, Ministry of Science and Technology, India | BT/RLF/Re-entry/50/2020 | Jogender Singh |
| Science and Engineering Research Board | SRG/2020/000022 | Jogender Singh |

The funders had no role in study design, data collection, and interpretation, or the decision to submit the work for publication.

### Author contributions

Gokul G, Conceptualization, Data curation, Formal analysis, Investigation, Methodology; Jogender Singh, Conceptualization, Data curation, Formal analysis, Funding acquisition, Investigation, Methodology, Project administration, Supervision, Visualization, Writing - original draft, Writing - review and editing

### Author ORCIDs

Jogender Singh  http://orcid.org/0000-0002-7947-0405

### Decision letter and Author response

Decision letter https://doi.org/10.7554/eLife.76021.sa1
Author response https://doi.org/10.7554/eLife.76021.sa2

# Additional files

## Supplementary files

• Transparent reporting form

## Data availability

The whole-genome sequence data for JSJ1-JSJ12 have been submitted to the public repository, the Sequence Read Archive, with BioProject ID PRJNA786771. All data generated or analyzed during this study are included in the manuscript and supporting files. Source Data files have been provided for Figure 1, Figure 2, Figure 3, Figure 4, Figure 5, Figure 6, Figure 1—figure supplement 1, Figure 2—figure supplement 1, Figure 3—figure supplement 2, Figure 3—figure supplement 3, Figure 3—figure supplement 4, and Figure 6—figure supplement 1.

The following dataset was generated:

| Author(s) | Year | Dataset title | Dataset URL | Database and Identifier |
|---|---|---|---|---|
| Indian Institute of Science Education and Research Mohali | 2021 | C. elegans DTT-resistant mutants | https://www.ncbi.nlm.nih.gov/bioproject/PRJNA786771/ | NCBI BioProject, PRJNA786771 |

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
