## [Editor Report]

DTT is thought to lead to cellular stress through its effect on the ER milieu and on disulfide bond formation in the ER. The authors show that DTT also affects the methionine cycle and that SAM depletion in fact leads to significant DTT toxicity in *C. elegans*, challenging the current model.

---

## [Decision Letter]

**Decision letter after peer review:**

Thank you for submitting your article "Dithiothreitol causes toxicity in *C. elegans* by modulating the methionine-homocysteine cycle" for consideration by *eLife*. Your article has been reviewed by 3 peer reviewers, including Martin Sebastian Denzel as the Reviewing Editor and Reviewer #1, and the evaluation has been overseen by David Ron as the Senior Editor.

Essential revisions:

To support the conclusions made in the paper, the reviewers agreed on a number of essential revisions:

1. SAM levels should be quantified to support the idea that they are in fact depleted after DTT exposure.

2. A genetic characterization of drm-1. It would be important to test if a genetic KO phenocopies the point mutants and if overexpression sensitize to DTT.

3. Published work needs to be cited more accurately and comprehensively. Particularly the experiments with B12/DTT were published before and should be cited in the figure legend.

4. Alternative plausible hypotheses explaining the DTT effects need to be considered.

*Reviewer #1 (Recommendations for the authors):*

To better support the conclusions, I would like to suggest the following key experiments.

1. To put the effect of DTT on SAM in context and to make comparisons in other systems possible, it would be important to measure the absolute SAM concentration in worms with and without DTT treatment, and after methionine supplementation.

2. To address a broad biological significance of the finding that DTT treatment first leads to toxicity through SAM depletion and only at higher concentrations to ER stress it would be important to perform the key experiments in cultured mammalian cells.

3. What is the effect of 5mM DTT on hsp-4::GFP expression/UPR signaling?

*Reviewer #2 (Recommendations for the authors):*

1. drm-1 should be fully characterized. It is a point mutation, therefore could be a loss, partial or gain of function allele. In addition, identification of the methylation pathway the drm-1 gene works in critical to bring together aspects of this story.

2. The data from this paper is largely self reported (*C. elegans* larval stage) or low magnification images. More quantitive assays should be used (for example, expression of adult collagens such as col-19), or higher magnification images of vulval development shown.

3. GFP images should be quantitated.

4. Data from the Walhout lab should be directly cited in the figure legend and in the same paragraph that discuss the results.

5. Direct mechanistic links from the action of DTT to the Met/SAM cycle are needed for the conclusions of the paper to be supportable. There are multiple types of experiments that might be acceptable here.

*Reviewer #3 (Recommendations for the authors):*

1. The authors conclude on L305 that "By upregulating the expression of SAM-dependent methyltransferase gene drm-1, DTT leads to the depletion of SAM that results in toxicity". No experiment has shown that drm-1 activation causes lower SAM – this is merely implied from the supplementation data. The authors should provide support of that model and measure SAM levels with and without DTT and with and without B12, as well as in the drm-1 mutant. Such molecular evidence of actual SAM depletion is critical to the main conclusion of the paper – that SAM levels and their depletion are a major driver of DTT toxicity.

2. If the authors' model is correct that DTT-driven drm-1 induction is the source of developmental delay due to SAM depletion, they should cross their overexpression transgene into the WT background; the prediction would be that this should cause developmental delay on its own. Alternatively, they could use a stronger intestinal promoter to overexpress drm-1. Again, is in the above point, the authors should also quantify SAM in these worms.

3. One idea that emerges from the data provided here is that drm-1 may be a SAM-dependent enzyme that produces PC; this might be important in the condition of DTT-induced ER stress as it might allow synthesizing more ER membrane, allowing for accumulation of misfolded proteins. It would be great if the authors could test whether the drm-1 mutant has lower levels of PC.

4. The role of thiol agents should be expanded beyond DTT. The authors could readily test whether the mechanism described here also protects against toxic concentrations of other agents, e.g. test B12/methionine/choline and/or drm-1 rescue of toxic concentration of NAC.

5.The authors showed that drm-1 is induced by DTT, and so were two of its orthologs, R08E5.1, R08F11.4 (but not K12D9.1). To examine whether protection against DTT is is specific to drm-1, the authors should use mutants in the other three genes to test whether they, too, block toxic effects of DTT (one might predict the former two should, the latter maybe not, because it is not induced by DTT). RNAi might work as well, there is now an option to do RNAi in OP50. The screen did not identify alleles in these genes, and while this doesn't rule out their importance, it is somewhat compelling and suggests that drm-1 is key here, so these experiments are not essential.

[Editors' note: further revisions were suggested prior to acceptance, as described below.]

Thank you for resubmitting your work entitled "Dithiothreitol causes toxicity in *C. elegans* by modulating the methionine-homocysteine cycle" for further consideration by *eLife*. Your revised article has been evaluated by David Ron (Senior Editor) and a Reviewing Editor.

The manuscript has been improved but there are some remaining issues that need to be addressed, as outlined below:

1. SAM data should be displayed in absolute values and in the same graph so that it's possible to understand the effect of *rips-1* mutation on SAM.

2. For the choline rescue experiment, it was noted that the concentrations used were unusually high. It would be important to show rescue of a *sams-1* mutant with choline.

3. From the data it appears that knockdown of all 4 methyltransferases results in some rescue, so perhaps they are all involved? This requires no experimentation, just a change in the wording of the interpretation.

---

## [Author Response]

Essential revisions:To support the conclusions made in the paper, the reviewers agreed on a number of essential revisions:1. SAM levels should be quantified to support the idea that they are in fact depleted after DTT exposure.

We thank the reviewers for suggesting this critical experiment. We have quantified the SAM levels in the N2 and *rips-1(jsn11)* animals with and without exposure to 10 mM DTT. We observed that DTT indeed reduces SAM levels in N2, but not *rips-1(jsn11)* animals. The revised manuscript reads (lines 274-283): “The genetic and methionine supplementation experiments suggested that SAM depletion could be the primary cause of DTT toxicity. To establish that DTT indeed resulted in the depletion of SAM, we quantified SAM levels in animals with and without exposure to DTT. The exposure of the wild-type N2 animals to DTT resulted in a significant reduction in its SAM levels (Figure 5F). To study whether SAM depletion by DTT depended on the gene *rips-1*, we quantified SAM levels in *rips-1(jsn11)* animals. Surprisingly, *rips-1(jsn11)* animals had significantly increased SAM levels upon exposure to DTT (Figure 5G). It is unclear why DTT exposure resulted in the increased SAM levels in *rips-1(jsn11)* animals. Nevertheless, these results establish that DTT exposure depletes SAM levels in N2, but not *rips-1(jsn11)* animals.”

2. A genetic characterization of drm-1. It would be important to test if a genetic KO phenocopies the point mutants and if overexpression sensitize to DTT.

We would like to highlight that four of the *rips-1* alleles isolated in our genetic screen are likely loss-of-function mutations (3 premature stop codons and one splice acceptor mutations). We have emphasized this information in the manuscript (lines 181-183): “Three of the isolated alleles of *rips-1*, jsn4, jsn11, and jsn12 had premature stop codon mutations and one allele (jsn8) had a splice acceptor mutation (Figure 3A). All of these mutations are expected to result in a loss-of-function of the gene.” The loss-of-function nature of *rips-1* mutations is further confirmed by a complete rescue of the resistance of *rips-1*(jsn11) animals to DTT toxicity upon transgenic expression of wild-type copies of *rips-1* (Figure 3F).

In addition, as suggested by the reviewers, we have now characterized the effects of *rips-1* knockdown by RNAi on the toxic effects of DTT. Knockdown of *rips-1* in the N2 animals resulted in a drastic improvement in their development in the presence of DTT (Figure 3—figure supplement 3A-B). Moreover, we have also characterized the impact of overexpression of *rips-1* on the sensitivity to DTT toxicity. Animals overexpressing *rips-1* exhibited enhanced sensitivity to DTT toxicity (Figure 3—figure supplement 3C-D). Taken together, these studies establish that the loss-of-function of *rips-1* imparts DTT resistance while its overexpression sensitizes the animals to DTT toxicity.

3. Published work needs to be cited more accurately and comprehensively. Particularly the experiments with B12/DTT were published before and should be cited in the figure legend.

We have now cited the earlier work on *acdh-1p::GFP* and bacterial diets both in the figure legend and the text that describes the results on *acdh-1p::GFP*.

4. Alternative plausible hypotheses explaining the DTT effects need to be considered.

We have added a new paragraph in the discussion to provide a hypothesis for explaining the effects of DTT. The revised manuscript reads (lines 377-392): “One question that arises from the current study concerns the substrate for the SAM-methyltransferase RIPS-1. Because DTT results in the induction of *rips-1* expression, one possibility is that DTT is the substrate for RIPS-1, and *rips-1* expression is induced in response to DTT as a defense mechanism. This hypothesis is supported by studies that showed that multiple thiol reagents, including DTT and β-ME, could be methylated by SAM and SAM-dependent methyltransferases (Bremer and Greenberg, 1961; Coiner et al., 2006; Maldonato et al., 2021). The methylation of external thiols could be a defense response for some organisms. For example, the dithiol metabolite gliotoxin produced by Aspergillus fumigatus inhibits the growth of other fungi, including A. niger, and triggers a defense response in A. niger resulting in upregulation of SAM-dependent methyltransferases (Dolan et al., 2014; Manzanares-miralles et al., 2016; Owens et al., 2015). Methylation of gliotoxin by SAMdependent methyltransferases confers A. niger protection against the dithiol metabolite. Incidentally, the same methyltransferases that are required for the defense mechanism against gliotoxin are induced by DTT (MacKenzie et al., 2005; Manzanares-miralles et al., 2016). Therefore, upregulation of methyltransferases might be a general defense response against multiple thiol reagents.”

Reviewer #1 (Recommendations for the authors):To better support the conclusions, I would like to suggest the following key experiments.1. To put the effect of DTT on SAM in context and to make comparisons in other systems possible, it would be important to measure the absolute SAM concentration in worms with and without DTT treatment, and after methionine supplementation.

We thank the reviewer for suggesting this important experiment. We have quantified the SAM levels in the N2 and *rips-1(jsn11)* animals with and without exposure to 10 mM DTT. We observed that DTT indeed lowers SAM levels in N2, but not *rips-1(jsn11)* animals. The revised manuscript reads (lines 274-283): “The genetic and methionine supplementation experiments suggested that SAM depletion could be the primary cause of DTT toxicity. To establish that DTT indeed resulted in the depletion of SAM, we quantified SAM levels in animals with and without exposure to DTT. The exposure of the wild-type N2 animals to DTT resulted in a significant reduction in its SAM levels (Figure 5F). To study whether SAM depletion by DTT depended on the gene *rips-1*, we quantified SAM levels in *rips-1(jsn11)* animals. Surprisingly, *rips-1(jsn11)* animals had significantly increased SAM levels upon exposure to DTT (Figure 5G). It is unclear why DTT exposure resulted in the increased SAM levels in *rips-1(jsn11)* animals. Nevertheless, these results establish that DTT exposure depletes SAM levels in N2, but not *rips-1(jsn11)* animals.”

2. To address a broad biological significance of the finding that DTT treatment first leads to toxicity through SAM depletion and only at higher concentrations to ER stress it would be important to perform the key experiments in cultured mammalian cells.

We thank the reviewer for this important suggestion. We believe that studying the impact of DTT on SAM and the methionine-homocysteine cycle in mammalian cells merits an in-depth investigation and is beyond the scope of the current manuscript.

3. What is the effect of 5mM DTT on hsp-4::GFP expression/UPR signaling?

We have added data on the effect of 5 mM DTT on hsp-4p::GFP expression. The revised manuscript reads (lines 314-318): “Next, we explored the role of ER stress in DTTmediated development retardation. First, we tested whether a lower concentration of DTT, which results in development retardation, would lead to ER stress. Exposure to 5 mM DTT resulted in the upregulation of hsp-4p::GFP to an intermediate level between the control and 10 mM DTT exposure levels (Figure 6—figure supplement 1A-B), indicating that lower concentrations of DTT also cause ER stress.”

Reviewer #2 (Recommendations for the authors):1. drm-1 should be fully characterized. It is a point mutation, therefore could be a loss, partial or gain of function allele. In addition, identification of the methylation pathway the drm-1 gene works in critical to bring together aspects of this story.

We would like to highlight that four of the *rips-1* alleles isolated in our genetic screen are likely loss-of-function mutations (3 premature stop codons and one splice acceptor mutations). We have emphasized this information in the manuscript (lines 181-183): “Three of the isolated alleles of *rips-1*, jsn4, jsn11, and jsn12 had premature stop codon mutations and one allele (jsn8) had a splice acceptor mutation (Figure 3A). All of these mutations are expected to result in a loss-of-function of the gene.” The loss-of-function nature of *rips-1* mutations is further confirmed by a complete rescue of the resistance of *rips-1(jsn11)* animals to DTT toxicity upon transgenic expression of wild-type copies of *rips-1* (Figure 3F).

In addition, we have now characterized the effects of *rips-1* knockdown by RNAi on the toxic effects of DTT. Knockdown of *rips-1* in the N2 animals resulted in a drastic improvement in their development in the presence of DTT (Figure 3—figure supplement 3AB). Moreover, we have also characterized the impact of overexpression of *rips-1* on sensitivity to DTT toxicity. Animals overexpressing *rips-1* exhibited enhanced sensitivity to DTT toxicity (Figure 3—figure supplement 3C-D). Taken together, these studies establish that the loss-of-function of *rips-1* imparts DTT resistance while its overexpression sensitizes the animals to DTT toxicity.

We have also added a discussion on the potential substrate of *rips-1*. The revised manuscript reads (lines 377-392): “One question that arises from the current study concerns the substrate for the SAM-methyltransferase RIPS-1. Because DTT results in the induction of *rips-1* expression, one possibility is that DTT is the substrate for RIPS-1, and *rips-1* expression is induced in response to DTT as a defense mechanism. This hypothesis is supported by studies that showed that multiple thiol reagents, including DTT and β-ME, could be methylated by SAM and SAM-dependent methyltransferases (Bremer and Greenberg, 1961; Coiner et al., 2006; Maldonato et al., 2021). The methylation of external thiols could be a defense response for some organisms. For example, the dithiol metabolite gliotoxin produced by Aspergillus fumigatus inhibits the growth of other fungi, including A. niger, and triggers a defense response in A. niger resulting in upregulation of SAMdependent methyltransferases (Dolan et al., 2014; Manzanares-miralles et al., 2016; Owens et al., 2015). Methylation of gliotoxin by SAM-dependent methyltransferases confers A. niger protection against the dithiol metabolite. Incidentally, the same methyltransferases that are required for the defense mechanism against gliotoxin are induced by DTT (MacKenzie et al., 2005; Manzanares-miralles et al., 2016). Therefore, upregulation of methyltransferases might be a general defense response against multiple thiol reagents.”

2. The data from this paper is largely self reported (*C. elegans* larval stage) or low magnification images. More quantitive assays should be used (for example, expression of adult collagens such as col-19), or higher magnification images of vulval development shown.

We thank the reviewer for this suggestion. However, most of the comparative development data reported in the manuscript have a great contrast in the development stages (for example, L1/L2 stages vs. L4/adult stages) which is readily observable without the need for development stage-specific reporters. We believe that the current resolution of data is adequate for the observed phenotypes.

3. GFP images should be quantitated.

We have quantified the GFP images. The quantification data is shown in Figure 3I, Figure 3—figure supplement 4B, Figure 4D, Figure 6B, and Figure 6—figure supplement 1B.

4. Data from the Walhout lab should be directly cited in the figure legend and in the same paragraph that discuss the results.

We have now cited the earlier work on *acdh-1p::GFP* and bacterial diets both in the figure legend and the text that describes the results on *acdh-1p::GFP*.

5. Direct mechanistic links from the action of DTT to the Met/SAM cycle are needed for the conclusions of the paper to be supportable. There are multiple types of experiments that might be acceptable here.

To provide a direct link between DTT and Met/SAM cycle, we have quantified the SAM levels in the N2 and *rips-1(jsn11)* animals with and without exposure to 10 mM DTT. We observed that DTT indeed reduces SAM levels in N2, but not *rips-1(jsn11)* animals. The revised manuscript reads (lines 274-283): “The genetic and methionine supplementation experiments suggested that SAM depletion could be the primary cause of DTT toxicity. To establish that DTT indeed resulted in the depletion of SAM, we quantified SAM levels in animals with and without exposure to DTT. The exposure of the wild-type N2 animals to DTT resulted in a significant reduction in its SAM levels (Figure 5F). To study whether SAM depletion by DTT depended on the gene *rips-1*, we quantified SAM levels in *rips-1(jsn11)* animals. Surprisingly, *rips-1(jsn11)* animals had significantly increased SAM levels upon exposure to DTT (Figure 5G). It is unclear why DTT exposure resulted in the increased SAM levels in *rips-1(jsn11)* animals. Nevertheless, these results establish that DTT exposure depletes SAM levels in N2, but not *rips-1(jsn11)* animals.”

Reviewer #3 (Recommendations for the authors):1. The authors conclude on L305 that "By upregulating the expression of SAM-dependent methyltransferase gene drm-1, DTT leads to the depletion of SAM that results in toxicity". No experiment has shown that drm-1 activation causes lower SAM – this is merely implied from the supplementation data. The authors should provide support of that model and measure SAM levels with and without DTT and with and without B12, as well as in the drm-1 mutant. Such molecular evidence of actual SAM depletion is critical to the main conclusion of the paper – that SAM levels and their depletion are a major driver of DTT toxicity.

We thank the reviewer for suggesting this important experiment. We have quantified the SAM levels in the N2 and *rips-1(jsn11)* animals with and without exposure to 10 mM DTT. We observed that DTT indeed reduces SAM levels in N2, but not *rips-1(jsn11)* animals. The revised manuscript reads (lines 274-283): “The genetic and methionine supplementation experiments suggested that SAM depletion could be the primary cause of DTT toxicity. To establish that DTT indeed resulted in the depletion of SAM, we quantified SAM levels in animals with and without exposure to DTT. The exposure of the wild-type N2 animals to DTT resulted in a significant reduction in its SAM levels (Figure 5F). To study whether SAM depletion by DTT depended on the gene *rips-1*, we quantified SAM levels in *rips-1(jsn11)* animals. Surprisingly, rips^-1^(jsn11) animals had significantly increased SAM levels upon exposure to DTT (Figure 5G). It is unclear why DTT exposure resulted in the increased SAM levels in *rips-1(jsn11)* animals. Nevertheless, these results establish that DTT exposure depletes SAM levels in N2, but not *rips-1(jsn11)* animals.”

2. If the authors' model is correct that DTT-driven drm-1 induction is the source of developmental delay due to SAM depletion, they should cross their overexpression transgene into the WT background; the prediction would be that this should cause developmental delay on its own. Alternatively, they could use a stronger intestinal promoter to overexpress drm-1. Again, is in the above point, the authors should also quantify SAM in these worms.

We thank the reviewer for another very important suggestion. We tested the sensitivity of *rips-1* overexpressing animals to DTT and indeed found them to be more sensitive to DTT. The revised manuscript reads (lines 216-218): “To establish that the toxic effects of DTT are indeed due to the upregulation of *rips-1*, we tested the effects of overexpression of *rips-1* on DTT sensitivity. Animals overexpressing *rips-1* exhibited enhanced sensitivity to DTT toxicity (Figure 3—figure supplement 3C-D).” We could not quantify SAM levels in the *rips-1* overexpressing animals as the array is extrachromosomal (resulting in both transgenic and non-transgenic progeny), and several thousands of worms are required to quantify SAM levels.

3. One idea that emerges from the data provided here is that drm-1 may be a SAM-dependent enzyme that produces PC; this might be important in the condition of DTT-induced ER stress as it might allow synthesizing more ER membrane, allowing for accumulation of misfolded proteins. It would be great if the authors could test whether the drm-1 mutant has lower levels of PC.

We thank the reviewer for this interesting conjecture. However, we observed that choline supplementation, which enhances PC synthesis, rescues DTT toxicity. Because SAM levels are critical for maintaining PC levels (in the absence of choline supplementation), it is likely that DTT leads to reduced levels of PC (supported by the rescue of DTT toxicity by choline supplementation). Therefore, if the *rips-1* mutants had lower levels of PC, they would have been more sensitive to DTT. However, because *rips-1* mutants are resistant to DTT, it is unlikely that *rips-1* is involved in PC synthesis.

4. The role of thiol agents should be expanded beyond DTT. The authors could readily test whether the mechanism described here also protects against toxic concentrations of other agents, e.g. test B12/methionine/choline and/or drm-1 rescue of toxic concentration of NAC.

We thank the reviewer for this important suggestion. We studied whether the thiol reagents, β-mercaptoethanol (β-ME) and NAC, caused toxicity via *rips-1*. Interestingly, we observed that β-ME, but not NAC, shares the toxicity pathway with DTT. These results are described in a new Results subsection and a new Figure (Figure 4). The revised manuscript reads (lines 263-272): “β-Mercaptoethanol, but not NAC, shares toxicity pathway with DTT We tested whether the other thiol reagents, β-mercaptoethanol (β-ME) and NAC, caused toxicity via *rips-1*. Interestingly, *rips-1(jsn11)* animals exhibited resistance to β-ME toxicity, but not NAC toxicity (Figure 4A-B). We studied the effect of these thiol reagents on the expression levels of *rips-1p::GFP*. While NAC exposure did not affect the expression of rips1p::GFP, β-ME resulted in its dramatic upregulation (Figure 4C-D). Finally, we studied the effects of vitamin B12 supplementation of the toxicities of β-ME and NAC. Vitamin B12 supplementation alleviated β-ME, but not NAC, toxicity (Figure 4E-F). These results suggested that β-ME shares the toxicity mechanism with DTT. On the other hand, NAC causes toxicity by a mechanism different from DTT.”

5.The authors showed that drm-1 is induced by DTT, and so were two of its orthologs, R08E5.1, R08F11.4 (but not K12D9.1). To examine whether protection against DTT is is specific to drm-1, the authors should use mutants in the other three genes to test whether they, too, block toxic effects of DTT (one might predict the former two should, the latter maybe not, because it is not induced by DTT). RNAi might work as well, there is now an option to do RNAi in OP50. The screen did not identify alleles in these genes, and while this doesn't rule out their importance, it is somewhat compelling and suggests that drm-1 is key here, so these experiments are not essential.

We studied the role of *rips-1* and *rips-1*-related methyltransferases in DTT toxicity by their RNAi-mediated knockdown. The revised manuscript reads (202-215): “Because DTT resulted in the upregulation of methyltransferases closely related to *rips-1*, we used RNAi knockdown to study the role of these methyltransferases in the toxic effects of DTT. As expected, knockdown of *rips-1* in the N2 animals resulted in a drastic improvement in their development in the presence of DTT (Figure 3—figure supplement 3A-B). Interestingly, knockdown of R08E5.1 resulted in a phenotype similar to *rips-1* knockdown. On the other hand, knockdown of R08F11.4 and K12D9.1 improved the development in the presence of DTT only marginally (Figure 3—figure supplement 3A-B). These results suggest that the *rips-1*-related methyltransferases might also be required for DTT toxicity. It is also possible that the knockdown of these methyltransferases results in the knockdown of *rips-1* by cross RNAi. The level of sequence similarity between *rips-1* and R08E5.1 (78 % identical) indicates that these genes would undergo complete cross RNAi (Rual et al., 2007). Because we did not recover mutants of any of the *rips-1*-related methyltransferases in our genetic screen in contrast to the twelve alleles of *rips-1*, the *rips-1*-related methyltransferases likely have only minor roles in DTT toxicity.”

[Editors' note: further revisions were suggested prior to acceptance, as described below.]

The manuscript has been improved but there are some remaining issues that need to be addressed, as outlined below:1. SAM data should be displayed in absolute values and in the same graph so that it's possible to understand the effect of *rips-1* mutation on SAM.

We thank the reviewer for these suggestions. We have displayed the SAM quantification data for both N2 and *rips-1(jsn11)* animals in the same graph (Figure 5F).

SAM is a very unstable molecule. The SAM standard that we purchased from Σ (#A7007) has this cautionary note: “This material is 80-90% pure when prepared, but is very unstable. As much as 10% purity loss per day at 25 °C has been noted.” Therefore, its absolute quantification may not be reliable. It is a standard practice to present SAM levels in relative values. Because for each experiment, all four of the samples (N2 and *rips-1(jsn11)* with and without 10 mM DTT) were prepared and processed simultaneously, the relative SAM quantification in these samples is reliable and sufficient to understand the effects of DTT on relative SAM levels.

2. For the choline rescue experiment, it was noted that the concentrations used were unusually high. It would be important to show rescue of a *sams-1* mutant with choline.

We have shown the rescue of *sams-1(ok2946)* animals with choline in Figure 5GH. While the rescue with choline is complete in N2 and metr-1(ok521) animals, it is only partial in *sams-1(ok2946)* animals. This suggested that while phosphatidylcholine is one important downstream product of SAM involved in attenuating DTT toxicity, other SAMrelated functions may also be involved in rescuing DTT toxicity. We have added these points in the manuscript (lines 295-301): “Supplementation of choline rescued the developmental defects in N2 and metr-1(ok521) animals on an *E. coli* OP50 diet containing 10 mM DTT (Figure 5G-H). The rescue of DTT toxicity was only partial in *sams-1(ok2946)* animals (Figure 5G-H). These results suggested that phosphatidylcholine is a major SAM product responsible for combating DTT toxicity. However, the partial rescue of DTT toxicity by choline supplementation in *sams-1(ok2946)* animals suggested that other SAM-related functions may also be involved in attenuating DTT toxicity.”

3. From the data it appears that knockdown of all 4 methyltransferases results in some rescue, so perhaps they are all involved? This requires no experimentation, just a change in the wording of the interpretation.

We thank the reviewer for this suggestion. Indeed, the knockdown of all 4 methyltransferases results in some rescue of DTT toxicity. This could be because of the following two reasons: Either all the 4 methyltransferases are involved in mediating DTT toxicity, or because of high sequence similarity among these methyltransferases, cross-RNAi takes place. We have now discussed both of these possibilities. We have added a new sentence in the manuscript (lines 215-216): “However, our studies do not rule out the involvement of the *rips-1*-related methyltransferases in DTT toxicity.” The whole paragraph on methyltransferases RNAi reads: “Because DTT resulted in the upregulation of methyltransferases closely related to *rips-1*, we used RNAi knockdown to study the role of these methyltransferases in the toxic effects of DTT. As expected, knockdown of *rips-1* in the N2 animals resulted in a drastic improvement in their development in the presence of DTT (Figure 3—figure supplement 3A-B). Interestingly, knockdown of R08E5.1 resulted in a phenotype similar to *rips-1* knockdown. On the other hand, knockdown of R08F11.4 and K12D9.1 improved the development in the presence of DTT only marginally (Figure 3— figure supplement 3A-B). These results suggest that the *rips-1*-related methyltransferases might also be required for DTT toxicity. It is also possible that the knockdown of these methyltransferases results in the knockdown of *rips-1* by cross RNAi. The level of sequence similarity between *rips-1* and R08E5.1 (78 % identical) indicates that these genes would undergo complete cross RNAi (Rual et al., 2007). Because we did not recover mutants of any of the *rips-1*-related methyltransferases in our genetic screen in contrast to the twelve alleles of *rips-1*, the *rips-1*-related methyltransferases likely have only minor roles in DTT toxicity. However, our studies do not rule out the involvement of the *rips-1*-related methyltransferases in DTT toxicity.”